# UMA: A Family of Universal Models for Atoms

Brandon M. Wood [1,*,†]    Misko Dzamba [1,*]    Xiang Fu [1,*]    Meng Gao [1,*]    Muhammed Shuaibi [1,*]

Luis Barroso-Luque [1]    Kareem Abdelmaqsoud [2]    Vahe Gharakhanyan [1]    John R. Kitchin [2]

Daniel S. Levine [1]    Kyle Michel [1]    Anuroop Sriram [1]    Taco Cohen [1]    Abhishek Das [1]

Ammar Rizvi [1]    Sushree Jagriti Sahoo [1]    Zachary W. Ulissi [1]    C. Lawrence Zitnick [1,†]

[1]FAIR at Meta        [2]CMU

## Abstract

The ability to quickly and accurately compute properties from atomic simulations is critical for advancing a large number of applications in chemistry and materials science including drug discovery, energy storage, and semiconductor manufacturing. To address this need, we present a family of Universal Models for Atoms (UMA), designed to push the frontier of speed, accuracy, and generalization. UMA models are trained on half a billion unique 3D atomic structures (the largest training runs to date) by compiling data across multiple chemical domains, e.g. molecules, materials, and catalysts. We develop empirical scaling laws to help understand how to increase model capacity alongside dataset size to achieve the best accuracy. The UMA small and medium models utilize a novel architectural design we refer to as mixture of linear experts that enables increasing model capacity without sacrificing speed. For example, UMA-medium has 1.4B parameters but only ∼50M active parameters per atomic structure. We evaluate UMA models on a diverse set of applications across multiple domains and find that, remarkably, a single model without any fine-tuning can perform similarly or better than specialized models. We are releasing the UMA code, weights, and associated data to accelerate computational workflows and enable the community to build increasingly capable AI models.

## 1 Introduction

Density Functional Theory (DFT) models the interaction of atoms from first principles through the estimation of their electronic structure. It serves as the foundation of modern computational chemistry and materials science, and has provided insights into many applications including drug discovery [70, 78], energy storage [73, 27, 72], and semiconductors [62, 76]. Despite DFT's widespread adoption, its considerable computational expense limits its usage.

Machine learning models offer the potential to accurately approximate DFT while being dramatically faster ($O(n)$ vs. $O(n^3)$ for DFT, where $n$ is the number of atoms); reducing computation time from hours to less than a second. Ideally, these models – referred to as Machine Learning Interatomic Potentials (MLIPs) – would generalize across the many tasks that utilize DFT. We refer to a "task" as a chemical domain (e.g. materials, catalysts, molecules) plus the specialized DFT settings required by domain. For each task, we can explore different applications of DFT, such as molecular dynamics, relaxations, vibrational analysis, and mechanical response. Training MLIPs that generalize across such tasks remains an open problem.

---

*Co-first Author

†Co-corresponding Author

39th Conference on Neural Information Processing Systems (NeurIPS 2025).

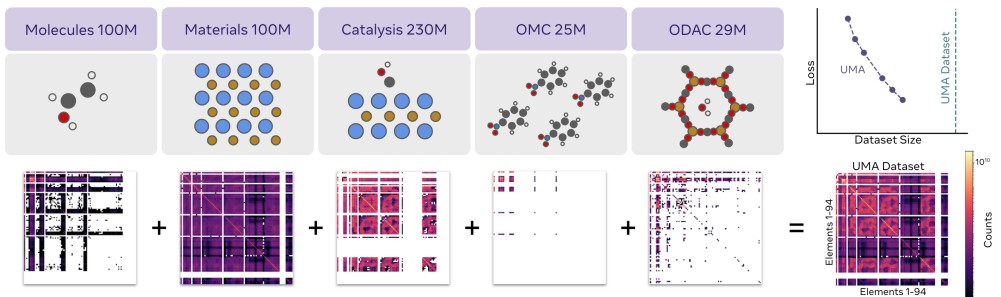

Figure 1: Visualization of the different datasets used for training. The 2D plots (bottom) illustrate the number of pairwise interactions contained in each dataset for every combination of elements. Note their combination covers nearly the entire chemical space with the exception of the radioactive elements. Model accuracies have improved with training dataset size (upper right), and this paper explores the limits of this scaling.

The scaling of datasets and model sizes has led to major breakthroughs in language and vision models, enabling their generalization across diverse data distributions and tasks [2, 24]. A similar scaling of atomic datasets is more challenging due to their computational cost, resulting in models typically being trained on smaller problem-specific datasets [57, 18, 17]. A potential solution is to pool data across tasks to allow for the creation of an exceptionally large dataset for training multi-task models. Recently, several large domain-specific datasets including catalysts [11, 72], materials [5, 67], and molecules [43] have been released that, when combined, total nearly half a billion atomic systems. This combined dataset defines a new paradigm in terms of the diversity of interactions and chemical environments (Figure 1), allowing for the learning of general-purpose models.

In this paper, we present a family of Universal Models for Atoms (UMA) designed to test the limits of accuracy, speed, and generalization for a single model across chemistry and materials science. The unprecedented amount of data (∼500M atomic systems, Figure 1) poses new challenges in balancing accuracy and efficiency with multi-task training. To explore this space, we develop empirical scaling laws, relating compute, data, and model size, to determine the model size required to fit the UMA dataset and to define compute-optimal and inference-optimal training strategies. To accommodate the growing demand for model capacity while maintaining speed, we introduce a novel Mixture of Linear Experts (MoLE) architecture that efficiently scales the model size without increasing inference times for applications such as Molecular Dynamics (MD). For efficient model training, we propose a novel two-stage training schedule that efficiently pre-trains a model that is then refined in the second stage to conserve energy at higher precision.

We demonstrate that UMA, without fine-tuning to specific tasks, performs similarly or better in both accuracy and inference-speed/memory-efficiency than specialized models on a wide-range of material, molecular and catalysis benchmarks. Highlights include state-of-the-art results on the popular Matbench Discovery leaderboard [59], a $25\%$ improvement in successful adsorption energy calculations for catalysis [42], and accuracy sufficient for practical applications (e.g. ligand-strain energy) in structure-based drug-design [43]. All code and data used for training UMA is publicly available. UMA model weights are available with a commercially permissive license (with some geographic and acceptable use restrictions).

## 2 Approach

An MLIP acts as a surrogate for DFT, and so it requires the same inputs as DFT: atom positions, their atomic numbers, and optionally spin and charge information. As outputs, MLIPs estimate the energy of an atomic structure from which other properties may be computed through the use of derivatives, such as per-atom forces, stress, etc. For many applications, such as molecular dynamics or performing relaxations, an MLIP is used to calculate the atomic forces to run simulations that can require thousands or even millions of iterations. For this reason, MLIPs must be both accurate and computationally efficient.

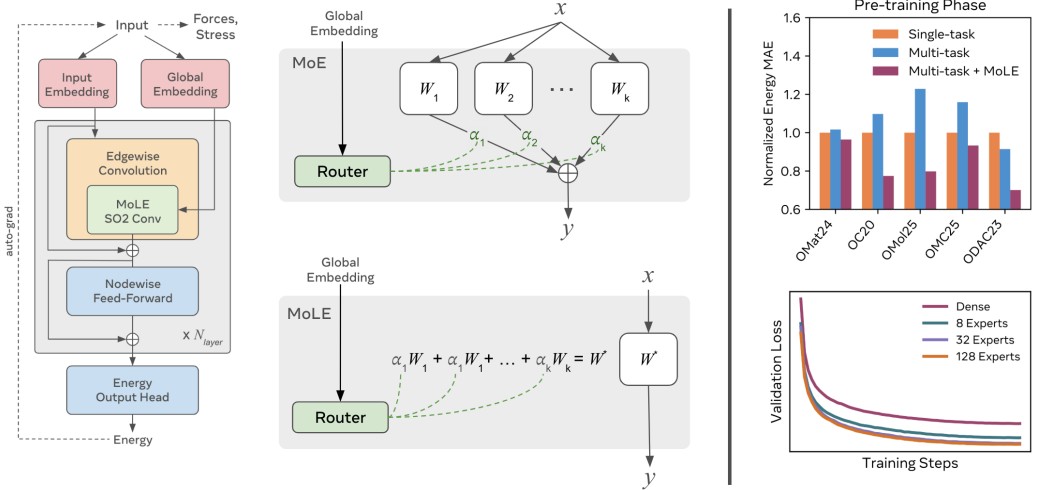

Figure 2: (left) Overview of UMA model architecture. The SO2 convolution is made up of a set of linear operations and each one of these operations is replaced with MoLE. (middle) Illustration of MoE and MoLE. The embedding used for routing, which estimates the expert weights $\alpha$, only depends on global information making it possible to merge before the model forward pass (middle, bottom), which has substantial benefits for applications that require long roll outs such as molecular dynamics. (right) Bar plot of UMA-S trained with MoLE for multi-task and without MoLE for single and multi-task. Note the MoLE model outperforms non-MoLE models. (right, bottom) Loss plots when varying the number of experts from 1 to 128 for UMA-S.

Table 1: Summary of UMA models.

| Model | Total Parameters | Active Parameters | Inferences per second for 1k Atoms | Max Atoms per 80GB GPU | Conservative |
|-------|------------------|-------------------|-----------------------------------|------------------------|--------------|
| UMA-S | 150M | 6M | 16 | 100k+ | ✓ |
| UMA-M | 1.4B | 50M | 3 | 10k+ | ✓ |
| UMA-L | 700M | 700M | 1.6 | 1k+ | ✗ |

Inference speed and max atoms measured on Nvidia H100 with a periodic system that has $\approx 50$ neighbors per atom within 6Å, see Appendix D

## 2.1 UMA: Universal Model for Atoms

In this paper, we describe a family of models that can be categorized into three sizes: small (S), medium (M), and large (L). A summary of the models can be found in Table 1. Additional details on the different versions of UMA can be found in Appendix E.1. All the models have different accuracy and speed attributes and as a result are best suited for distinct use cases. UMA-S aims to strike a balance between accuracy and efficiency and is suitable for computationally intensive applications such as long molecular dynamics simulations. The most general-purpose model of the family is UMA-M, which is more accurate and can be used as a DFT surrogate for relaxations or vibrational analysis as well as for molecular dynamics. UMA-L is a highly accurate DFT surrogate and is intended as a proof-of-principle to help understand scaling behavior and to demonstrate the limits of what is currently feasible.

We begin by providing a basic introduction to the UMA architecture. Next, we highlight our use of Mixture of Linear Experts (MoLE), an efficient way to scale capacity and add flexibility for training models across different DFT datasets and tasks. Lastly, we present an overview of the training procedure used for UMA models.

## 2.2 Architecture

The UMA architecture is based on eSEN [21], an equivariant graph neural network, with a number of important modifications to enable the model to efficiently scale and handle additional inputs, such as total charge and spin, and the DFT settings desired for emulation. The standard eSEN architecture

takes 3D atomic positions and atomic numbers as inputs and returns the total energy, per-atom forces, and, optionally, stress. The network operates by updating a spherical harmonic node embedding for each atom through a series of message passing layers. The embeddings are initialized based on the atom's atomic number. Messages are passed between neighboring atoms that are less than a predefined distance (6 Å) from each other. Each message passing layer consists of an edge-wise block followed by a node-wise feed-forward block with residual connections. Normalization is performed after each layer. The central component of the edgewise block is the eSCN convolution [54]. After message passing, there is a single task-independent node-wise feed-forward block to predict the outputs.

### 2.2.1  Charge, Spin, and DFT Task Inputs

In addition to the inputs handled by eSEN, we expand the model to incorporate information about the system's total charge, total spin multiplicity, and the DFT task. The charge and spin are indicated using an integer value for each and allow the MLIP to model structures which may be electrically charged (have extra or missing electrons) or have unpaired electrons. The DFT task indicates which of the five training dataset DFT settings the model should attempt to replicate. A single dataset is specified and a random vector corresponding to it is fed into the network. This is necessary since different datasets use different plane-wave or localized orbital DFT calculators (VASP [39, 38] and ORCA [50]) and levels of theory.

We introduce a new embedding to UMA models that enables the inclusion of charge, spin, and DFT task. Each of these inputs generates an embedding with the same dimension as the number of spherical channels used, which is concatenated and then fed through a 1-layer feed-forward network. The result is added to the node embeddings at each layer for the spherical harmonics coefficients of degree 0 ($L = 0$). This embedding is also used to compute the global embedding used for MoLE routing, which is described below.

### 2.3  Mixture of Linear Experts

A common approach to learning general-purpose models is to increase the amount and diversity of their training data. As datasets increase in size, scaling laws suggest model sizes must also be scaled to optimally reduce losses [24, 2]. However, this presents a tradeoff, which may lead to accurate models that are too computationally expensive to use in practical applications. One method to improve these trade-offs is to use Mixture of Experts [32, 31, 47] (MoEs) to increase the number of model parameters while minimizing the additional computational cost. In an MoE model, the outputs of a block are calculated by a set of experts, each with their own individual set of weights. Typically, a sparse weighted combination of their outputs is combined and passed to the next block. This approach has been shown to work well for LLMs to improve both their efficiency and their ability to generalize [19, 33, 60, 15].

The application of MoEs to MLIPs requires additional considerations. Contrary to language modeling, estimating the potential energy surface is a regression task whose outputs should vary smoothly to ensure energy conservation [21]. In addition, the estimated forces should be equivariant to rotation. This implies that the use of experts should not introduce discontinuities on the energy surface, and care must be taken to ensure equivariance is maintained. Another practical consideration for MLIPs is that the task and set of atoms is commonly held constant during long simulations. Ideally, an MoE approach for MLIPs would take advantage of this to improve efficiency in sequential inference applications.

We propose using a Mixture of Linear Experts (MoLE) for MLIPs. An MoLE combines a set of linear experts [46, 31]:

$$y = \sum_k \alpha_k \left( W_k x \right) \tag{1}$$

where each expert $k$ has a set of weights $W_k$ and contribution $\alpha_k \in [0, 1]$. While this was one of the original approaches proposed for MoEs [31], MoEs that use sparse sets of non-linear experts which compete for attention [32] are much more commonly used in modern networks [19, 60, 15, 63]. However, MoLEs offer distinct advantages for MLIPs. First, the MLIP can learn functions that share information and vary smoothly between tasks by encouraging the dense use of all experts without enforcing sparseness. Second, since MoLE is a mixture of linear experts, they maintain

rotational equivariance when used within the eSCN convolution [54]. Finally, if the expert weights are only dependent on time-invariant global information such as element composition, the network weights may be precomputed before running simulations [46, 75]. The precomputed weights $W^*$ are calculated by moving the summation in Equation 1, which results in MoE inference times that are similar to non-MoE models:

$$y = W^*x \quad \text{where} \quad W^* = \left( \sum_k \alpha_k W_k \right) \tag{2}$$

The contribution $\alpha_k$ of each expert is calculated as a function of system-level features, including element composition, charge, spin, and task information. Embeddings are calculated for each of these properties, concatenated, and passed through a 3-layer MLP followed by a softmax to estimate $\alpha_k, \sum_k \alpha_k = 1$. We specifically exclude information when calculating $\alpha$'s that may vary during the course of relaxations or MD simulations, such as relative atom positions or other neighborhood information.

## 2.4 Training Procedure

Training models on large datasets across numerous tasks is challenging due to the computational resources required. This is especially true for conservative models, which require an additional backward pass to calculate forces or stress. To improve training efficiency, we implemented a two-stage approach described in detail in the supplementary material (Appendix A). As proposed by [21, 7], we train a model that directly predicts forces in the first stage. In the second stage, we remove the force head and fine-tune the model to predict conserving forces and stresses using auto-grad. This approach takes advantage of the faster training enabled by direct models, while providing energy conservation and smooth potential energy landscapes required by many applications.

Several other novel improvements were made to each stage's training to further improve efficiency. Half-precision is critical to training efficiently on modern GPUs. Similar to other domains such as LLM training, we found that BF16 is significantly more stable compared to FP16 with automatic mixed precision. However, unlike LLMs, MLIPs are significantly more sensitive to numerical precision; we observed that BF16 alone will degrade accuracy by as much as $20 - 50\%$ depending on the task and the property being computed . We found that pre-training with BF16 and switching to FP32 for fine-tuning recovers the loss in accuracy. Finally, training the models with a large number of MoLE experts is challenging due to memory constraints. We make further optimization to help bound and amortize memory usage (Appendix A). In addition, models were trained with a combination of graph parallelism [66] and fully-sharded data parallelism for large MoLE layers and activation checkpointing (Appendix A). When combined, this allows us to reliably scale up model training up to 10B total parameters.

The energies for each dataset can vary significantly due to the use of different DFT settings, which makes multi-task training difficult. To account for this, an energy referencing scheme was employed as described in Appendix A.6. This approach allowed for the use of a single energy head across all tasks without the need for task-specific heads.

## 3 Datasets

To train a model capable of generalizing across DFT tasks, an ideal training dataset would include materials, molecules, and their interactions. The Open Molecules 2025 (OMol25) [43] and Open Materials 2024 (OMat24) [5] datasets cover both molecules and materials respectively. The Open Catalyst 2020 (OC20) [11] and OpenDAC 2025 (ODAC25) [68] datasets are useful for modeling the interactions of molecules and materials. Finally, the Open Molecular Crystals 2025 (OMC25) [23] dataset specializes in modeling the interaction between molecules in periodic structures. In combination, these datasets contain close to 500 million training examples with over 30 billion atoms. As visualized in Figure 1, the combination of these datasets contains nearly all pairwise interactions between atoms of different elements. We focused on large datasets, because of the challenges of mixing different DFT tasks and since the inclusion of much smaller datasets was unlikely to significantly impact the model weights.

While all DFT calculations yield energy and force labels by determining ground-state electronic structures, different chemical systems require domain-specific DFT settings. For instance, the OMat24

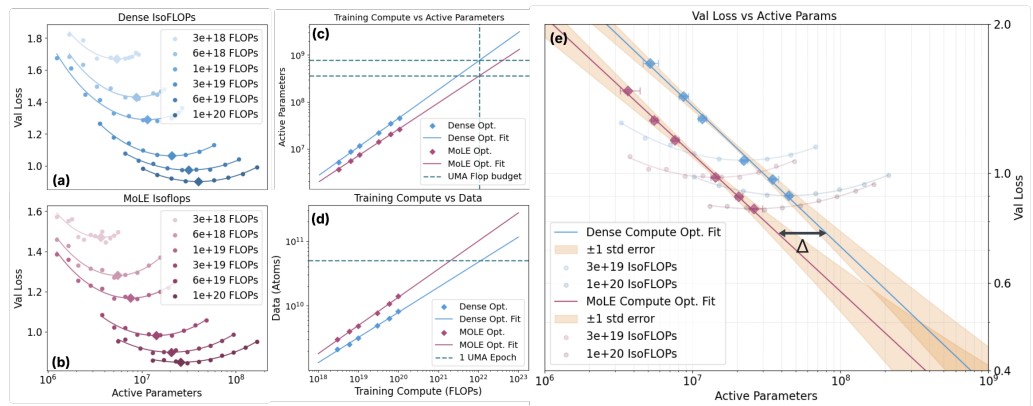

Figure 3: Empirical scaling measurements of dense (blue) vs. MoLE (red) model architectures. FLOPs vs. validation loss for (a) dense and (b) MoLE (8-expert) models. Experiment sets are performed by holding FLOPs constant and varying model size and training data. Diamonds represent the compute optimal frontier. (c) Training compute vs. parameters. Vertical green dotted line represents our estimated training budget of $O(10^{22})$ FLOPs, horizontal dotted lines are corresponding dense and MoLE model sizes. (d) Compute vs. dataset size (atoms) with the green dotted line representing the training FLOPs required for 1 epoch of UMA training data (50B atoms) (e) Overlay of dense vs. MoLE compute optimal frontiers from (a-b) and the fitted power law of validation loss as a function of parameters. A compute optimal MoLE model with $\Delta \approx 2.5\times$ fewer active parameters can achieve an equivalent loss. Fitting details and parameters are described in Appendix C.

dataset uses the PBE functional and is generated with the plane-wave code VASP, whereas OMol25 employs the $\omega$B97M-V functional with the localized orbital code ORCA. Designing a single model that performs well across such diverse settings is a non-trivial challenge. Additionally, the datasets differ in size and information content, so we adjust their sampling ratios: 4 for OMat24 and OMol25, 1 for OC20 and ODAC25, and 2 for OMC25. Further details on the datasets and our unified modeling strategy can be found in Appendix B and A respectively.

## 4 Results

### 4.1 Model and Data Scaling

In various domains such as language, empirical power-law relationships have been successfully used [28, 24, 35] to predict the optimal model size and quantity of training data given a fixed compute budget. However, for modeling MLIPs, no models have been trained on the size of UMA's dataset and very few models have been trained to >100M parameters [5]; we do not know whether such relationships exist or whether the models will continue to improve with data and parameters. In this study, we show that UMA models do show log-linear scaling behavior as seen in other domains in the FLOP ranges we tested, indicating that greater model capacity is required to fit the UMA dataset. We used these scaling relationships to select the appropriate model size for our dataset and demonstrate the advantage of MoLE over the dense architecture.

Following [28, 24, 35], we first examine the behavior of training compute in FLOPs $C$ as a function of model size $N$ and dataset size $D$. To determine this, we used total validation loss (computed in the same way as training loss) as the observable quantity. We trained a series of models by sweeping the model parameter size while fixing the training compute budget in FLOPs. This is done by varying the amount of training data for each model, i.e., smaller models receive more training data, while larger models receive less. This is performed for both the dense and MoLE (8 expert) models classes, producing "Iso-FLOPs" curves Figure 3(a,b). The minima of the Iso-FLOPs represent the compute optimal model, or the best achievable loss for the given compute budget (diamonds in Figure 3).

**Optimal model size.** We fit the Iso-FLOPs minima using power laws [35, 28] to estimate the optimal model and dataset sizes given a fixed compute budget (detailed in Appendix C). Figure 3(c,d)

indicates that the dense and MoLE models display convincing log-linear scaling behavior at the scales of our experiments ($10^{18} - 10^{20}$ FLOPs). Given that our training dataset was approximately 50 billion atoms (1 epoch) and our estimated compute budget for pretraining was $O(10^{22})$ FLOPs (extrapolated from our preliminary training runs), Figure 3(c) suggests the largest compute optimal dense model we should train is $\sim$ 700M parameters corresponding to the UMA-L. In practice, we used the compute optimal model size as a starting point and continued training beyond the compute optimal regime to produce lower losses while minimizing parameter sizes. However, we observed that training on too many epochs can lead to overfitting and significantly deviate from log-linear scaling behavior, prompting us to keep our training within 2-3 epochs.

**Dense vs MoLE.** If we combine Figures 3(a,b) and fit the Iso-FLOPs minima as a function of loss (Figure 3(e)), we can compare the model sizes needed by the dense and MoLE models to achieve a fixed loss. Effectively, an optimal MoLE model can achieve the same loss as an optimal dense model that is $\Delta$ times larger, e.g., $\Delta \approx 2.5 \pm 0.2$ for UMA-M. We observe that this advantage is reduced at larger model sizes; training a 700M active parameter (8 expert - 5.6B parameter total) MoLE model had marginal improvement over the dense version. This effect can be seen in overlaid IsoFLOPs in Figure 3(e). As parameters increase, dense and MoLE performance converge. We hypothesize that we are limited by our dataset's size, regardless of MoLE or dense architectures.

## 4.2 Multi-task vs. Single-Task

To explore the benefits of MoLE models for multi-task training, we compare UMA-S (with MoLE) to non-MoLE dense models trained for single/multi-task in Figure 2 (right, top). In this limited parameter regime, multi-task models trained without MoLE always perform worse than single-task (ST) models. With the use of MoLE, UMA-S can achieve results comparable to task specialized small models. We also experimented with varying the number of experts for UMA-S in Figure 2 (right, bottom). A significant improvement in loss is observed when moving from 1 expert to 8. A smaller gain is found with 32 experts, and the additional gain from 32 to 128 experts is negligible. In our experiments, UMA-S uses 32 experts. For large models, multi-task training offers benefits even without MoLE, as shown in Appendix F, where multi-task models avoid overfitting and generally achieve better performance.

## 4.3 Inference Efficiency

The ability to increase the length scale (number of atoms) and time scale of molecular dynamics (MD), relaxations and other types of long running simulations is critical to our design. Leveraging our MoLE strategy, we can pre-merge all the weights and pay no additional penalty in inference time or memory when running a long time-scale simulations on a single system. Here we show that UMA models, despite having very large parameter counts and achieving SOTA accuracy across many domains, are extremely competitive in simulation settings in both speed and raw number of atoms that can fit in memory compared to specialized single-domain models. For example, the UMA-S can simulate 1000 atoms at 16 steps per second (1.4 ns-per-day) and fit system sizes up 100,000 or more atoms in memory on a single 80GB GPU. Lastly, our models are designed to be scaled with multi-GPU parallel inference using the Graph Parallelism strategy [66] described in training; giving the potential to provide orders of magnitude speed-ups when running simulations at the 100k+ atoms scale and unlock new science that is not possible today. We will leave multi-GPU inference outside the scope of this paper.

## 4.4 Evaluation

Our evaluation contains two main components: (1) a diverse set of held-out test splits (Table 2) and (2) a suite of practically important benchmarks (Table 3). To cover all chemical domains studied: materials, catalysis, molecules, molecular crystals, and metal–organic frameworks (MOFs), we propose new test sets and benchmarks, in addition to established benchmarks in existing literature. We highlight selected results below, with a more detailed discussion in Appendix E.

**Materials.** In Table 2, we report the test-set performance on two OOD test sets: WBM [74] and high entropy alloy (HEA). The HEA test set is introduced in this work (Appendix E.2). UMA-M performs on-par with SOTA task specialized models, such as eSEN-30M-OMat and eqV2-OMat. The degraded performance of UMA-L on energy of the HEA dataset (Table 11) suggests overfitting,

Table 2: Test MAE results on held out test splits for materials [59], catalysis [11], molecules [43], molecular crystals [23] and MOFs [67]. All energies are in meV, forces are in meV/Å and stresses are in meV/Å$^3$. Results for UMA are compared against the SOTA literature results. Target accuracies for practical utility are provided as an approximate guide for reference.

| Model | Materials WBM Energy/Atom | Forces | Stress | HEA Energy/Atom | Forces | Stress | Catalysis ID Ads. Energy | Forces | OOD-Both Ads. Energy | Forces | Molecules OOD-Comp Energy/Atom | Forces | PDB-TM Energy/Atom | Forces | Molecular crystals OMC-Test Energy/Atom | Forces | Stress | MOFs OOD-L/T Ads. Energy | Forces |
|---|---|---|---|---|---|---|---|---|---|---|---|---|---|---|---|---|---|---|---|
| **UMA** | | | | | | | | | | | | | | | | | | | |
| UMA-S-1.1 | 20.2 | 62.8 | 4.4 | 24.9 | 83.7 | 3.5 | 51.5 | 24.1 | 68.8 | 30.7 | 0.95 | 8.64 | 0.84 | 15.47 | 1.03 | 5.04 | 0.93 | **289.9** | 13.3 |
| UMA-M-1.1 | 18.2 | 50.7 | 4.2 | 21.9 | 69.0 | 3.5 | **31.8** | 15.5 | **45.5** | 20.2 | **0.74** | **5.44** | **0.50** | **10.14** | **0.84** | **2.83** | **0.90** | 294.1 | 10.3 |
| **Literature** | | | | | | | | | | | | | | | | | | | |
| eSEN-30M-OMat [21] | 16.2 | 49.6 | 4.1 | **20.0** | 59.5 | 3.2 | - | - | - | - | - | - | - | - | - | - | - | - | - |
| eqV2-OMat [5] | **14.9** | **46.3** | **3.6** | 20.3 | **47.0** | **2.7** | - | - | - | - | - | - | - | - | - | - | - | - | - |
| eqV2-OC20 [44] | - | - | - | - | - | - | 149.1 | **11.6** | 306.5 | **15.7** | - | - | - | - | - | - | - | - | - |
| GemNet-OC20 [22] | - | - | - | - | - | - | 163.5 | 16.3 | 343.3 | 23.1 | - | - | - | - | - | - | - | - | - |
| eqv2-ODAC [67] | - | - | - | - | - | - | - | - | - | - | - | - | - | - | - | - | - | 316.0 | **7.2** |
| eSEN-sm-cons. [43] | - | - | - | - | - | - | - | - | - | - | 1.35 | 7.39 | 0.83 | 12.72 | - | - | - | - | - |
| eSEN-S-OMC [23] | - | - | - | - | - | - | - | - | - | - | - | - | - | - | 1.05 | 5.39 | 0.94 | - | - |
| **Target** | | | | | | | | | | | | | | | | | | | |
| Practical Utility | 10-20 | - | - | 10-20 | - | - | 100 | - | 100 | - | 1-3 | - | 1-3 | - | 1-3 | - | - | 100 | - |

Table 3: Evaluation results on Matbench-Discovery [59], MDR phonon [45], elastic tensor [16, 34], and AdsorbML benchmarks [42]. Results are also provided for a diverse set of molecule [43] and molecular crystal [29, 23] benchmarks. NVE MD [21] tests whether energy conservation is observed when running the model for molecular dynamics. SOTA results from literature are reported where available. Additionally, for the materials evaluations, UMA models were fine-tuned on MPtrj [17] and sAlex [61, 5] to be consistent with the benchmark's DFT settings.

| Model | Materials Matbench [59] F1 | RMSD | MAE [eV/atom] | $\kappa_{srme}$ [56] | Phonons [45] $\omega_{max}$ [K] | Free Energy [kJ/mol] | Elasticity [16, 34] $G_{vrh}$ [GPa] | $K_{vrh}$ [GPa] | NVE MD [21] Conserve | Catalysis AdsorbML [42] Success Rate | Molecules OMol25 [43] Ligand-strain [meV] | PDB-pocket [meV] | Dist-SR [meV] | Dist-LR [meV] | NVE MD [21] Conserve | Molecular Crystals CSP Targets [29] Lattice Energy [kJ/mol] | Kendall Rank | RMSD [Å] |
|---|---|---|---|---|---|---|---|---|---|---|---|---|---|---|---|---|---|---|
| **UMA** | | | | | | | | | | | | | | | | | | |
| UMA-S-1.1 | 0.913 | 0.064 | 0.020 | 0.204 | 18.82 | 5.48 | 9.47 | 5.16 | ✓ | 66.80% | 4.86 | 127.7 | 20.4 | 194.7 | ✓ | **2.13** | **0.86** | **0.13** |
| UMA-M-1.1 | **0.929** | **0.061** | **0.018** | 0.176 | **14.81** | **3.87** | 8.57 | 4.78 | ✓ | 72.25% | 2.96 | **76.8** | 15.5 | 138.1 | ✓ | 3.24 | 0.82 | 0.14 |
| **Literature** | | | | | | | | | | | | | | | | | | |
| eSEN-30M-OAM [21] | 0.925 | 0.061 | 0.018 | **0.170** | 15.00 | 4.00 | 9.13 | 5.73 | ✓ | - | - | - | - | - | - | - | - | - |
| ORB v3 [58] | 0.905 | 0.075 | 0.024 | 0.210 | - | - | - | - | - | - | - | - | - | - | - | - | - | - |
| SevenNet-MF-ompa [36] | 0.901 | 0.064 | 0.021 | 0.317 | - | - | - | - | - | - | - | - | - | - | - | - | - | - |
| GRACE-2L-OAM [8] | 0.880 | 0.067 | 0.023 | 0.294 | - | - | - | - | - | - | - | - | - | - | - | - | - | - |
| MACE-MPA-0 [6] | 0.852 | 0.073 | 0.028 | 0.412 | - | - | - | - | - | - | - | - | - | - | - | - | - | - |
| eqv2-OC20 [42] | - | - | - | - | - | - | - | - | - | 60.80% | - | - | - | - | - | - | - | - |
| GemNet-OC20 [42] | - | - | - | - | - | - | - | - | - | 54.88% | - | - | - | - | - | - | - | - |
| eSEN-sm-cons. [43] | - | - | - | - | - | - | - | - | - | - | 4.66 | 147.3 | 21.6 | 197.0 | ✓ | - | - | - |
| eSEN-S-OMC [23] | - | - | - | - | - | - | - | - | - | - | - | - | - | - | - | 6.18 | 0.74 | 0.18 |

and demonstrates the test set's challenging nature. In Table 3, we report benchmark results on the prediction of materials' thermodynamic stability, thermal conductivity, and vibrational properties. Note for these results, the UMA models are fine-tuned on the MPTrj [17] and sAlex [61, 5] to be consistent with the DFT settings of the public benchmarks. On the popular benchmark Matbench-Discovery (MBD) [59], UMA-M achieves the highest F1 score to date. All UMA models of different sizes show strong performance, which suggests that further scaling of model sizes is unlikely to result in better performance on the MBD benchmark. The conserving UMA models (S and M) also excel at phonon-related properties, which are reflected by metrics including $\kappa_{\mathrm{SRME}}$, free energy MAE, and shear/bulk elasticity MAE.

**Catalysis.** We evaluate our models on two established catalysis benchmarks: OC20 S2EF [11] and AdsorbML [42]. These benchmarks focus on adsorption energy predictions – that is, the models predict the change in energy as a molecule, known as an adsorbate, comes in contact with a surface. Most previous models on the OC20 benchmark directly predict the adsorption energy given the adsorbate's position on the surface. Since our model predicts total energy, we make a total energy prediction on the same structure, and another on a clean surface without the adsorbate [1]. These two values are subtracted (along with the energy of the adsorbate in isolation) to predict the adsorption energy. As shown in Table 2, UMA models significantly outperform previous SOTA models on OC20 S2EF adsorption energy prediction – reducing the errors by around 80%. On the more realistic

benchmark of AdsorbML, which evaluates a model's capability to predict the global minimum adsorption energy, all UMA models outperform the previous SOTA (EquiformerV2). In particular, UMA-L achieves a 25% improvement in the success rate (Table 12).

**Molecules.** Initial UMA models were trained on a preview subset ($\approx 70\%$) of the OMol25 dataset, since portions of the dataset were still being calculated when UMA model training began (results in Appendix E). Both UMA-1 and UMA-1.1 were trained on the full OMol25 dataset. In Table 2, we report the prediction performance on two challenging test splits of OMol25: OOD-Comp and OOD PDB-TM. Our results show that UMA-S achieves performance comparable to the domain-specific eSEN-sm-cons, while UMA-M significantly outperforms both models. In Table 3, we evaluate ligand strain and pocket–ligand interaction energy, which are quantities relevant to structure-based drug discovery. In particular, for ligand strain (and, similarly, for conformers in Table 19), UMA-M achieves errors approaching those of the DFT reference, demonstrating its ability to act as a practical surrogate for these calculations. On the distance scaling benchmarks (short and long range) in Table 3, UMA-S slightly surpasses eSEN-sm-cons despite having the same receptive field.

**Molecular Crystals.** In contrast to materials, molecules, and catalysis, the development and use of universal MLIPs for molecular crystals has been relatively underexplored. In addition to the OMC25 test set, we evaluate our UMA's capability to accurately predict lattice energies, rank, and match structures of the most recent 7th Crystal Structure Prediction (CSP) Blind Test [29]. In all evaluations, UMA-S outperforms the task-specific eSEN-S-OMC baseline, which was trained on the OMC25 dataset [23]. The high accuracy of lattice energy predictions ($\leq 3$ kJ/mol) indicates that UMA can be a reliable substitute for DFT in many crystal structure prediction applications.

**Metal Organic Frameworks.** Computing the adsorption energy of $CO_2$ for a MOF sorbent is important for direct-air carbon capture applications and an established benchmark [67]. Similar to OC20, UMA models use total energy predictions to compute the adsorption energy. UMA models perform on par with previous SOTA models on the ID test set (Appendix E) while achieving the best performance on the hardest ODAC OOD test set (Table 2), suggesting improved generalization.

# 5 Related Work

## 5.1 Universal MLIPs

The field of MLIPs has been rapidly improving due in part to the availability of larger datasets. A recent example in the materials space is the release of the Alexandria [61] and OMat24 [5] datasets, which quickly led to improved performance on the Matbench-Discovery leaderboard [59]. While MLIPs trained on these materials datasets are often referred to as "universal" because they include nearly all the elements in the periodic table [12, 6, 20, 51, 48, 77, 53], they may not generalize well to other domains such as molecules [37] or surfaces [11]. One reason for this is the chemical, structural, and elemental distribution shifts between materials and other domains such as molecules. Another reason is the level of DFT theory that is considered accurate differs between domains, e.g. PBE for materials and $\omega$B97M-V for molecules. As we demonstrate, training across domains can lead to unified representations and help generalization.

While there have been a number of promising works exploring training MLIPs across multiple domains, it remains an open challenge to demonstrate high-accuracy zero-shot performance. One approach is to pre-train a model using a large corpus of data (> 100M) and fine-tuning for specific tasks [65, 79]. The fine-tuned models are shown to perform significantly better than models trained from scratch. Nevertheless, removing the need for specialization would make these models substantially more useful. Recent studies suggest that achieving zero-shot performance across two DFT tasks may be possible [36, 64].

## 5.2 Scaling Relations

Empirical scaling laws provide a wealth of insight into the relationships between compute, data, and model size. In LLMs, scaling laws motivated the community to train on more tokens with larger models because performance improvements became predictable [28, 24, 35]. Additionally, scaling relations help practitioners decide on how to best (compute-optimal) allocate resources such as dataset and model size. These types of relations have been explored in MLIPs to compare the asymptotic

scaling behavior of different model architectures and training paradigms. For example, [10] showed that equivariant models have different scaling behavior compared to non-equivariant models.

# 6    Limitations

While UMA represents a significant step forward, there are still limitations and areas for improvement. Currently, the UMA models are limited in their ability to model long-range interactions. Our small and medium models use a standard MLIP cutoff distance of 6Å, the actual receptive field is much larger due to message passing, but this can present a problem for inputs where sets of atoms are separated by more than 6Å. For example, if an adsorbate starts at 7Å from a catalyst's surface the model views these as two independent non-interacting structures. Improvements may also be made in how charge and spin are incorporated into the model [17]. Currently, each discrete charge or spin uses a separate embedding, which limits its ability to generalize to unseen spins or charges.

Our work aims to accelerate scientific discovery in chemistry and materials science using MLIPs; however, potential misuses exist, and we train on datasets designed for applications beneficial to society to mitigate these risks.

# 7    Discussion and Conclusion

We explore techniques for training MLIPs across diverse DFT tasks using nearly 500 million training examples gathered over numerous datasets. By taking advantage of scaling relationships between dataset size, model capacity and loss, we are able to train models that are competitive with–or better than–task specialized models. Naive approaches to scaling model parameters would result in an accurate but slow model. To address this, we introduced Mixture of Linear Experts, a method to increase model capacity while maintaining inference efficiency. Among our family of models, UMA-S offers a favorable balance between speed and accuracy, capable of performing MD simulations of 1,000 atoms at 1.4ns per day on a single 80GB GPU.

We evaluated UMA across a wide-range of benchmarks for materials, molecules, catalysts, molecular crystals, and metal organic frameworks, demonstrating strong performance across all. For well established benchmarks, such as AdsorbML and Matbench Discovery, we achieve new state-of-the-art results. Looking ahead, the development of more challenging benchmarks will be crucial for driving progress in the field.

Our findings suggest that a single model can achieve sufficient accuracy for practical research applications across a broad spectrum of chemistry and materials science applications. This paves the way for universal MLIPs and opens new opportunities for atomic simulations at scale.

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

## Appendix Table of Contents

# A  Training Details and Hyperparameters

## A.1  Two-stage training

While conservative models have been found to provide reliable performance in diverse physical property prediction tasks [21], the backward pass required for force/stress prediction significantly increases inference costs. Models with direct force prediction are much more efficient, and have been found to be effective as a pre-training strategy to save compute when subsequently fine-tuned as a conservative model [21, 7]. The UMA-S and UMA-M both follow this procedure. In addition, we introduce low-precision training, max-atom batching, and max neighbor switching to further enhance the scalability and efficiency of our training process. These novel strategies are discussed in turn below. Detailed hyper-parameters are summarized in Table 4.

**Precision.** For pretraining, we used BF16 numerical format, commonly used in training LLMs but uncommon in MLIP models due high numerical precision requirements. In our experiments, we found that BF16 is significantly more stable than AMP-FP16 (automatic mixed precision) especially in our multi-modal setting where data distributions can vary dramatically, frequently causing gradient and loss spikes that would destabilize AMP training. However, it suffers an accuracy drop compared to AMP-FP16 and FP32. We found the degradation can be nearly completely recovered after a very small number of finetuning steps in FP32 (<1% of data).

**Max-atom Batching.** Due to the large differences in system topology and number of atoms/edges per system, using a fixed number of systems as batch size is infeasible. Instead we chose to use a max-atom batching scheme where we randomly pack batches that contain as close to an upper bound (max atoms) as possible without going over to guarantee an upper bound on memory usage.

**Max Neighbors.** For training efficiency purposes, we use a significantly smaller number of neighbors per atom during pretraining and found that it has no effect on the final performance, energy conservation, and smoothness properties of the model after finetuning with effectively "infinite" max neighbors.

## A.2  Parallelisms

Although our models are designed with inference efficiency in mind, training models with a large number of MoLE experts is memory intensive, In particular, for the finetuning stages, a combination of infinite neighbors, FP32 precision, and autograd forces puts significant constraints on memory and training speed. We used a combination three parallelism training techniques summarized as follows:

- Graph parallelism (GP) [66]: Partitioning graphs across GPUs during message passing layers is used when scaling up to a large number of atoms at large model sizes. Graph partitioning is only used within a node with a fixed graph parallel rank size (2 or 4) during conservative fine-tuning stages.
- Fully-sharded data parallel (FSDP): We use the Pytorch FSDPv1 implementation on MoLE expert layers only for models with a total parameter count exceeding 1B during conservative fine-tuning when memory is scarce. Parameters are sharded within a node and replicated across nodes.
- Distributed Data Parallel (DDP): We use the standard PyTorch DDP implementation, with modifications for per-atom loss averaging and compatibility with graph parallelism.

Furthermore, we leveraged Pytorch's Distributed Checkpointing framework to ensure saving and loading extremely large checkpoints is efficient and stable across different node configurations. Exponential moving average (EMA) is used for stable validation performance.

## A.3  Model

**One model for all tasks.** UMA is designed for multi-task learning under diverse DFT settings. For two inputs with exactly the same atomic numbers and positions, the DFT labels will be different when different DFT settings are used. Such DFT settings include the level of theory and system total charge/spin. These task specifications are global information of the atomic system, and we process them through initial embedding layers. In this paper, five levels of theories are involved –

Table 4: Summary of main training-related hyper-parameters for the pre-training and fine-tuning stages. These hyper-parameters are shared among model sizes.

| Hyper-parameter | Pre-training | Finetuning |
|---|---|---|
| Precision | BF16 | FP32 |
| Radius cutoff Å | 6 | 6 |
| Max neighbors | 30 | 300 |
| Force prediction | Direct | Autograd |
| Stress prediction | None | Autograd |
| Optimizer | AdamW | AdamW |
| Learning rate scheduling | Cosine | Cosine |
| Maximum learning rate | $8 \times 10^{-4}$ | $4 \times 10^{-4}$ |
| Warmup epochs | 0.01 | 0.01 |
| Warmup factor | 0.2 | 0.2 |
| Gradient clipping norm threshold | 100 | 100 |
| Model EMA decay | 0.999 | 0.999 |
| Weight decay | $1 \times 10^{-3}$ | $1 \times 10^{-3}$ |
| Energy loss coefficient | 10 | 20 |
| OMol energy loss coefficient | 30 | - |
| Force loss coefficient | 30 | 2 |
| Stress loss coefficient | - | 1 |

Table 5: Hyper-parameters for UMA models of different sizes.

| Hyper-parameters | UMA-S | UMA-M | UMA-L |
|---|---|---|---|
| Number of MoLE experts | 32 | 32 | Dense |
| Number of layer blocks | 4 | 10 | 16 |
| Maximum degree $L_{\max}$ | 2 | 4 | 6 |
| Maximum order $M_{\max}$ | 2 | 2 | 2 |
| Number of channels $N_{\text{channel}}$ | 128 | 128 | 256 |
| Number of radial basis functions | 64 | 128 | 256 |
| Global batch size (atoms) | 88k | 44k | 44k |
| Number of pre-training steps | 1.68M | 2.08M | 2.58M |
| Number of fine-tuning steps | 1M | 545k | 350k |

OMat24, OC20, OMol25, OMC25, and ODAC25 all use different DFT levels of theory. OMol25 further contains systems with non-neutral charge/spin.

For this iteration of the UMA models, these global information are embedded as follows:

Furthermore, to use a single model for all tasks, it is crucial to normalize the labels such that targets from different datasets fall into similar numerical ranges. We specifically design a referencing scheme that brings the diverse datasets to a similar level, detailed in Appendix B. The model hyperparameters are shown in Table 6.

### A.4   MPtrj and sAlex Fine-tuning

For materials evaluations, UMA models were fine-tuned on the MPTrj [17] and sAlex [61, 5] datasets to ensure consistent DFT settings. The fine-tuning procedure is the same as eSEN-30M-OAM as documented in [21].

### A.5   Training Compute Resources

The resources used to train UMA models are described in Table 7.

### A.6   Referencing and Normalization

While each dataset used in this work comes with its own specific set of DFT settings, we wanted a referencing scheme that provides a way to make energy magnitudes comparable across datasets. We

Table 6: Hyper-parameters for UMA models of different sizes.

| Hyper-parameters | UMA-S | UMA-M | UMA-L |
|---|---|---|---|
| Number of MoLE experts | 32 | 32 | Dense |
| Number of layer blocks | 4 | 10 | 16 |
| Maximum degree $L_{\max}$ | 2 | 4 | 6 |
| Maximum order $M_{\max}$ | 2 | 2 | 2 |
| Number of channels $N_{\text{channel}}$ | 128 | 128 | 256 |
| Number of radial basis functions | 64 | 128 | 256 |
| Global batch size (atoms) | 88k | 44k | 44k |
| Number of pre-training steps | 1.68M | 2.08M | 2.58M |
| Number of fine-tuning steps | 1M | 545k | 350k |

Table 7: Training Times for UMA models.

| Model | Stage | GPUs in Parallel | Training Days | GPU-Type |
|---|---|---|---|---|
| UMA-S | Direct Pre-train | 128 | 5 | H200 140GB |
| UMA-S | Conserve Fine-tune | 256 | 5 | H200 140GB |
| UMA-M | Direct Pre-train | 128 | 14 | H200 140GB |
| UMA-M | Conserve Fine-tune | 256 | 14 | H200 140GB |
| UMA-L | Direct Pre-train | 128 | 25 | H100 80GB |
| UMA-L | Stress Fine-tune | 128 | 4 | H100 80GB |
| UMA-L | FP32 Fine-tune | 128 | 2 | H100 80GB |

do this through a "heat of formation" (HOF) reference that is applied to the energies:

$$E_{ref} = E_{DFT} - \sum_i^N \left[ E_{i,DFT} - \Delta H_{f,i} \right]$$

Where $E_{DFT}$ corresponds to the total DFT energy of the system, $i$ is the atom number, $N$ is the total number of atoms in the system, $E_{i,DFT}$ is the DFT energy of an isolated atom $i$ in a box, and $\Delta H_{f,i}$ is the heat of formation of atomic number $i$ as taken directly from Mendeleev [49]. $E_{i,DFT}$ was calculated using the DFT settings for each of the unique datasets in this work. Additionally, we apply a linear reference to the above energies to help with the convergence and training stability of our models. We follow the same protocol as described in the OC22 Appendix [72].

We use a custom normalization $x' = \frac{x-\mu}{\sigma}$ for all targets (energy, forces, stress), where the shift term $\mu = 0$ and the scale term $\sigma =$ Force root mean square (RMS). For the combined dataset the Force RMS is computed as the weighted average (based on the number of systems) of all individual dataset Force RMS values.

## B  Training Data

A summary of the five datasets used to train UMA is shown in Table 8. In total, the dataset has 459 million training examples, containing up to 350 atoms. The average number of atoms varies based on the dataset, from 19 for OMat24 to 178 for ODAC25.

Table 8: Overview of the five datasets used to train UMA. For each dataset various statitiscs are provided alongside the sampling ratio used for training.

| Dataset | Domain | Training Size | Labels | # Elements | Avg Size | Force RMS | Sampling ratio |
|---|---|---|---|---|---|---|---|
| OMat24 | Materials | 100,824,585 | E,F,S | 89 | 19 | 2.83 | 4 |
| OMol25 | Molecules | 75,889,983 | E,F | 83 | 52 | 0.985 | 4 |
| OC20++ | Catalysis | 229,054,043 | E,F | 56 | 77 | 0.624 | 1 |
| OMC25 | Molecular Crystals | 24,870,226 | E,F,S | 12 | 130 | 0.103 | 2 |
| ODAC25 | MOFs | 28,517,826 | E,F | 70 | 178 | 0.046 | 1 |
| **Total** | | 459,156,663 | | | | | |

### B.1 Materials

The field of inorganic bulk materials is moving at an incredibly fast pace. Here we train on the Open Materials (OMat24) dataset (100M) [5], one of the largest and most diverse datasets in the community. All DFT calculations from this domain were run with VASP [39, 38, 40, 41] and used the PBE [55] functional. Due to the differences in psuedopotential version and different pseudopotentials for certain elements in OMat24 and Materials Project [5] calculation settings used for the data in most third party benchmarks, finetuning was also performed on MPtrj [17] and subsampled Alexandria (sAlex) [61] to ensure consistent evaluation on the materials benchmarks.

### B.2 Molecules

The community has seen dozens of molecular datasets spanning different scales for a variety of applications []. However, the varying levels of DFT theory and quality makes it challenging to unify under a single model. The release of the Open Molecules 2025 (OMol25) dataset [43] helps address this by providing the largest single dataset (100M+) spanning 80+ elements covering metal-complexes, biomolecules, electrolytes, and several existing datasets under a single, high-quality level of theory. All DFT calculations were performed using Orca [50] ($\omega$B97M-V/def2-TZVPD). At the time of training, only 75M samples from OMol25 were available for use, and we refer to this as OMol-preview. Splits were constructed to ensure that this snapshot of the dataset is consistent with the full dataset release. All ablations and results were trained with this OMol-preview, unless stated otherwise. Released models, however, will be retrained with the full OMol25 dataset to ensure the best models are accessible by the community.

### B.3 Catalysis

The Open Catalyst (OC20) dataset [11] provides the largest adsorbate+surface dataset in the community. OC20 enumerates 1M+ unique surface + adsorbate combinations, spanning 55 elements, and runs local geometry optimizations. Here, we train on the OC20 All (133M), MD (38M), and Rattled (17M) datasets. Unlike prior work, we also leverage OC20's clean surface data (14M) since models here are trained on total energies. One limitation of OC20 is that it only contains single adsorbates that interact with a surface. To address this, we introduce the OC20-Multi-Adsorbate (mAds) dataset (22M) to better capture coverage effects and adsorbate-adsorbate interactions. All DFT calculations were performed using VASP [39, 38, 40, 41] with the RPBE exchange-correlation functional functional [26].

### B.4 Molecular Crystals

The most recent 7th Crystal Structure Prediction (CSP) Blind Test organized by Cambridge Crystallographic Data Center (CCDC) demonstrated the effectiveness of tailored machine learning interatomic potentials (MLIPs) in predicting, filtering, and ranking molecular crystal structures [30, 29]. However, despite the widespread applications of molecular crystals, there has been limited focus on developing universal MLIPs for molecular crystals, mostly because of the scarcity of publicly available datasets. Currently, publicly available datasets of molecular crystals are scarce, with at most 60,000 materials represented [3, 9]. To address this data gap, we use the Open Molecular Crystals (OMC25) dataset, which comprises 25 million molecular crystal structures. The dataset includes multiple relaxation trajectories of various molecular crystal packings generated with Genarris [71] starting from organic molecules from the OE62 dataset [69]. The dataset includes 12 elements (all elements from OE62, excluding Li, As, Se, and Te) and the maximum number of atoms is capped at 300. The dataset is computed using VASP [39, 38, 40, 41] with the PBE exchange-correlation functional [55] and D3 dispersion correction [25]. The OMC25 dataset, along with in-depth details on data and polymorph structure generation, will be released in an upcoming publication [23].

### B.5 Metal-organic frameworks (MOFs)

The Open Direct Air Capture 2025 (ODAC25) dataset represents the largest MOF dataset (78M) for DAC applications [68]. ODAC25 extends the earlier ODAC23 [67] dataset, and is derived from the CoREMOF [13, 14] dataset. ODAC25 focuses on the adsorption and co-adsorption of $CO_2$ and $H_2O$ in MOFs among other adsorbates, which is critical for DAC performance evaluation. This work uses

the subset of ODAC25 that overlaps with ODAC23, where all DFT-computed adsorption energies have been upgraded to a higher k-points sampling grid density, providing improved accuracy over ODAC23 (29M calculations). All DFT calculations were performed using VASP [39, 38, 40, 41] with the PBE exchange-correlation functional [55] and D3 dispersion correction [25].

## C  Scaling Laws Methods

The scaling law experiments were only performed on pretraining; due to compute constraints, we did not study the effect on finetuning with energy conservation. We used an 8-expert MoLE version of the model with the UMA-M settings ($l_{max} = 4$, $m_{max} = 2$, and $n_{neighbors} = 30$) for consistency.

### C.1  FLOP counting

We approximate our training FLOPs by

$$C(N, D) \approx \kappa N D, \tag{3}$$

where $N$ is the total number of parameters in the network, and $D$ is the number of inputs (tokens for LLMs and atoms or edges for MLIPs) computed on. This relationship holds for any network that is dominated by linear layers where $\kappa$ indicates how many times a parameters is re-used on an input. For a single forward pass of a linear network, $\kappa = 2$. A full training cycle for such a network with a single backward pass brings $\kappa = 6$ as we need to compute the gradient with respect to both the parameters and inputs. Thus, for most LLMs $\kappa = 6$ FLOPs/parameter/token [35] for a single forward and backwards pass where $D$ is in units of tokens.

For our edge-based SO2 equivariant networks [54], $\kappa$ is a function of $l_{max}$ and $m_{max}$ spherical harmonic orders and the number of edges per atom $n_{neighbors}$. For the scaling law experiments, we used $l_{max} = 4$, $m_{max} = 2$, and $n_{neighbors} = 30$ which corresponds to the settings of UMA-M model, resulting in $\kappa \approx 270$ FLOPs/parameter/atom (or 9 FLOPs/parameter/edge) per training step, which can be computed or experimentally determined. As long as the number of input edges and parameters are sufficiently large (which holds for UMA models), this flop approximation holds as contributions from all other operators are marginal. We also verified this assumption with direct FLOP counting in our model code.

Overall, parameter reuse is significantly higher in Equivariant-GNNs compared to LLMs, and hence the flop count is 2-3 orders of magnitude higher for a similar parameter-sized LLM network.

### C.2  Compute optimal fits

The compute optimal model and dataset sizes can then be fitted to power laws [35, 28]:

$$\begin{aligned} \log(N^*(C)) &= \alpha \log(C) + A \\ \log(D^*(C)) &= \beta \log(C) + B \end{aligned} \tag{4}$$

Where $C$ is the compute in FLOPs described in Appendix C.1. $N^*(C)$ represents the optimal model size (in parameters) as a function of compute, and $D^*(C)$ represents the optimal dataset size (in units of atoms). $N^*(C)$ is determined by finding the minima of fitted parabolas for each Isoflop curve. The 10% and 90% percentile bootstrap errors are shown in Figure 3(e). $\alpha$ and $\beta$ are the scaling coefficients w.r.t. model size and dataset size respectively. $A$ and $B$ are offset constants of the fit. Fit coefficients and bootstrap errors are shown in Table 9.

### C.3  Fitting loss vs. parameters

To understand the minimum achievable loss for dense vs MoLE models we can fit the more general parameterized ansatz of $L(N, D)$ proposed by [35].

$$\tilde{L}(N, D) = \hat{E} + \frac{\hat{A}}{N^{\hat{\alpha}}} + \frac{\hat{B}}{D^{\hat{\beta}}} \tag{5}$$

This maps the power law coefficients from Equation 5 to those in Equation 4 by using $\alpha = \frac{\hat{\beta}}{\hat{\alpha}+\hat{\beta}}$ and $\beta = \frac{\hat{\alpha}}{\hat{\alpha}+\hat{\beta}}$. Here we can either minimize the $\tilde{L}(N,D)$ directly by fitting the 5 parameters $\hat{E}, \hat{A}, \hat{B}, \hat{\alpha}, \hat{\beta}$ with a iterative minimization procedure such as LBFGS [28, 10] or by examining the loss as a power relationship of $N^*$ using

$$\log \tilde{L}(N^*) = \hat{\alpha}\log(N^*) + \gamma \tag{6}$$

where $\gamma = log([1 + \frac{\hat{\alpha}}{\hat{\beta}}]\hat{A})$ with $\hat{E} \approx 0$. We found both methods yielded similar results but the minimization of $\tilde{L}$ was more sensitive to the choice of hyperparameters.

Table 9: Power Law Coefficients determined from fitting Equations 4 and 6. Error bounds are determined by bootstrap sampling 1000 times and taking the 10th and 90th percentile values, quoted in brackets.

| Parameter | Dense | MoLE |
|---|---|---|
| $\alpha$ | 0.61 (0.57,0.65) | 0.56 (0.49,0.59) |
| $\beta$ | 0.39 (0.35, 0.43) | 0.44 (0.39, 0.43) |
| $A$ | -4.5 (-3.8, -5.3) | -3.8 (-2.56, -4.65) |
| $B$ | 3.6 (2.9, 4.4) | 2.9 (1.6, 3.7) |
| $\hat{\alpha}$ | -0.29 (-0.27, -0.31) | -0.25 (-0.2, -0.3) |
| $\gamma$ | 2.16 (2.02, 2.34) | 1.82 (1.61, 2.12) |

# D  Inference

For inference benchmarking we use a periodic fcc carbon system with lattice constant $a = 3.8$Å. This results in a fixed density of approximately 50 edges per atom within 6Å. For UMA models, we use a combination of torch.compile, cuda graphs and pre-merged MoLE experts for inference speed. For large number of atoms ($> 1000$), we use edge-based activation checkpointing to trade off memory for some speed, allowing us to fit 100k+ atoms for the UMA-S into memory. We checked all our optimizations chosen does not degrade simulation accuracy, equivariance or energy conservation properties. While our benchmarks do not include graph generation, our internal CUDA based graph generation algorithm is very fast and decreases throughput by no more than $10\%$ even for the largest systems tested. In the case of non-MoLE merging, we found the inference speed was comparable but the parameters require more GPU memory to store.

Table 10: Single-GPU simulation speeds for energy-conservative models in *steps per second*: comparing conservative UMA models to the top two models (eSEN and OrbV3) on the Matbench Discovery leaderboard [59] and the MACE materials and molecules models. Benchmarks are run, excluding graph generation, on a single Nvidia H100 80GB GPU using FP32 (TF32-high precision) and torch compile when possible. Test systems are a standard periodic atomic system that have $\approx 50$ neighbors per atom with a 6Å cutoff. OOM indicates the model ran out of memory. Refer to Sec.D for more details.

| Atoms | UMA-S (6.6M) | UMA-M (50M) | eSEN-30M-OAM (30M) | Orb-v3 conservative-inf-omat (25M) | MACE-MPA-0 (9M) | MACE-OFF23-L (4.7M) |
|---|---|---|---|---|---|---|
| 100 | 44 | 21 | 8 | 77 | 38 | 89 |
| 1,000 | 16 | 3 | 1.7 | 30 | 24 | 20 |
| 10,000 | 1.6 | 0.2 | OOM | 3.7 | 2.9 | OOM |
| 50,000 | 0.2 | OOM | OOM | OOM | OOM | OOM |
| 100,000 | 0.1 | OOM | OOM | OOM | OOM | OOM |

For fair comparisons against other models, we used pytorch2.6.0, cuda12.4, python3.12 and TF-32 precision universally on a H100 80GB GPU. We use standard torch.compile settings whenever possible (only MACE-MPA-0 failed to compile). Different models have different radius cutoffs and

max neighbors settings. We made sure that all models was receiving roughly 50 neighbors per atom for the same number of atoms.

# E  Evaluations

**Section E.1** provides results for all versions of UMA.

**Sections E.2-E.6** provides additional results for the original version of UMA, which was trained on a preview subset of the OMol25 dataset.

Table 11: Test MAE results on held out test splits for materials [59], catalysis [11], molecules [43], molecular crystals [23] and MOFs [67]. All energies are in meV, forces are in meV/Å and stresses are in meV/Å$^3$. Results for UMA are compared against the SOTA literature results. Target accuracies for practical utility are provided as an approximate guide for reference.

| | Materials | | | | | | Catalysis | | | | Molecules | | | | Molecular crystals | | | MOFs | |
| | WBM Energy/Atom | Forces | Stress | HEA Energy/Atom | Forces | Stress | ID Ads. Energy | Forces | OOD-Both Ads. Energy | Forces | OOD-Comp Energy/Atom | Forces | PDB-TM Energy/Atom | Forces | OMC-Test Energy/Atom | Forces | Stress | OOD-L/T Ads. Energy | Forces |
| Model | | | | | | | | | | | | | | | | | | | |
|---|---|---|---|---|---|---|---|---|---|---|---|---|---|---|---|---|---|---|---|
| **UMA** | | | | | | | | | | | | | | | | | | | |
| UMA-S | 20.0 | 60.8 | 4.4 | 22.0 | 72.8 | 3.1 | 52.1 | 24.3 | 70.2 | 30.9 | 3.64 | 10.80 | 0.88 | 16.12 | 0.91 | 4.77 | 0.97 | 292.4 | 16.0 |
| UMA-M | 18.1 | 51.4 | 4.3 | 19.0 | 62.2 | 3.2 | 33.4 | 16.0 | 46.5 | 21.0 | 3.26 | 9.09 | 0.69 | 10.37 | 0.82 | 3.00 | 0.98 | 290.2 | 10.7 |
| UMA-L | 17.6 | 45.5 | 3.8 | 24.8 | 48.3 | 2.8 | 32.4 | 12.2 | 43.5 | 15.9 | 2.33 | 5.19 | 0.81 | 8.76 | 0.59 | 2.28 | 0.10 | 291.1 | 6.5 |
| UMA-S-1 | 19.4 | 62.2 | 4.5 | 21.9 | 73.5 | 3.1 | 53.1 | 24.5 | 70.4 | 31.2 | 0.96 | 8.25 | 0.93 | 15.56 | 0.93 | 5.15 | 1.01 | 287.1 | 13.6 |
| UMA-S-1.1 | 20.2 | 62.8 | 4.4 | 24.9 | 83.7 | 3.5 | 51.5 | 24.1 | 68.8 | 30.7 | 0.95 | 8.64 | 0.84 | 15.47 | 1.03 | 5.04 | 0.93 | 289.9 | 13.3 |
| UMA-M-1.1 | 18.2 | 50.7 | 4.2 | 21.9 | 69.0 | 3.5 | 31.8 | 15.5 | 45.5 | 20.2 | 0.74 | 5.44 | 0.50 | 10.14 | 0.84 | 2.83 | 0.90 | 294.1 | 10.3 |
| **Literature** | | | | | | | | | | | | | | | | | | | |
| eSEN-OMat [21] | 16.2 | 49.6 | 4.1 | 20.0 | 59.5 | 3.2 | - | - | - | - | - | - | - | - | - | - | - | - | - |
| eqV2-OMat [5] | 14.9 | 46.3 | 3.6 | 20.3 | 47.0 | 2.7 | - | - | - | - | - | - | - | - | - | - | - | - | - |
| eqV2-OC20 [44] | - | - | - | - | - | - | 149.1 | 11.6 | 306.5 | 15.7 | - | - | - | - | - | - | - | - | - |
| GemNet-OC20 [22] | - | - | - | - | - | - | 163.5 | 16.3 | 343.3 | 23.1 | - | - | - | - | - | - | - | - | - |
| eSEN-sm-cons. [43] | - | - | - | - | - | - | - | - | - | - | 1.35 | 7.39 | 0.83 | 12.72 | - | - | - | - | - |
| eSEN-S-OMC [23] | - | - | - | - | - | - | - | - | - | - | - | - | - | - | 1.05 | 5.39 | 0.94 | - | - |
| eqv2-ODAC [67] | - | - | - | - | - | - | - | - | - | - | - | - | - | - | - | - | - | 316.0 | 7.2 |
| **Target** | | | | | | | | | | | | | | | | | | | |
| Practical Utility | 10-20 | - | - | 10-20 | - | - | 100 | - | 100 | - | 1-3 | - | 1-3 | - | 1-3 | - | - | 100 | - |

Table 12: Evaluation results on Matbench-Discovery [59], MDR phonon [45], elastic tensor [16, 34], and AdsorbML benchmarks [42]. Results are also provided for a diverse set of molecule [43] and molecular crystal [29, 23] benchmarks. NVE MD [21] tests whether energy conservation is observed when running the model for molecular dynamics. SOTA results from literature are reported where available. For the materials evaluations, UMA models were fine-tuned on MPtrj [17] and sAlex [61, 5] to be consistent with the benchmark's DFT settings.

| | Materials | | | | | | | | | Catalysis | Molecules | | | | | Molecular Crystals | | |
| | Matbench [59] F1 | RMSD | MAE [eV/atom] | $\kappa_{syme}$ [56] | Phonons [45] $\omega_{max}$ [K] | Free Energy [kJ/mol] | Elasticity [16, 34] $G_{vrh}$ [GPa] | $K_{vrh}$ [GPa] | NVE MD [21] Conserve | AdsorbML [42] Success Rate | OMol25 [43] Ligand-strain [meV] | PDB-pocket [meV] | Dist-SR [meV] | Dist-LR [meV] | NVE MD [21] Conserve | CSP Targets [29] Lattice Energy [kJ/mol] | Kendall Rank | RMSD [Å] |
| Model | | | | | | | | | | | | | | | | | | |
|---|---|---|---|---|---|---|---|---|---|---|---|---|---|---|---|---|---|---|
| **UMA** | | | | | | | | | | | | | | | | | | |
| UMA-S | 0.916 | 0.064 | 0.020 | 0.203 | 17.59 | 5.00 | 8.54 | 4.96 | ✓ | 68.35% | 4.39 | 150.3 | 67.6 | 432.1 | ✓ | 2.695 | 0.82 | 0.12 |
| UMA-M | 0.930 | 0.061 | 0.018 | 0.195 | 13.91 | 3.39 | 8.40 | 4.76 | ✓ | 71.12% | 2.45 | 89.7 | 41.6 | 588.7 | ✓ | 2.664 | 0.81 | 0.13 |
| UMA-L | 0.928 | 0.065 | 0.018 | 0.671 | 78.50 | 18.20 | 20.56 | 14.48 | ✗ | 74.41% | 3.37 | 71.7 | 16.6 | 246.1 | ✗ | 2.488 | 0.84 | 0.12 |
| UMA-S | 0.916 | 0.064 | 0.020 | 0.203 | 17.59 | 5.0 | 8.54 | 4.96 | ✓ | 68.35% | 4.39 | 150.3 | 67.6 | 432.1 | ✓ | 2.70 | 0.82 | 0.12 |
| UMA-S-1 | 0.914 | 0.064 | 0.020 | 0.231 | 18.65 | 5.51 | 13.72 | 5.19 | ✓ | 64.85% | 5.19 | 138.3 | 23.8 | - | ✓ | 2.57 | 0.81 | 0.13 |
| UMA-S-1.1 | 0.913 | 0.064 | 0.020 | 0.204 | 18.82 | 5.48 | 9.47 | 5.16 | ✓ | 66.80% | 5.2 | 127.7 | 26.7 | 256.7 | ✓ | 2.13 | 0.86 | 0.13 |
| UMA-M-1.1 | 0.929 | 0.061 | 0.018 | 0.176 | 14.81 | 3.87 | 8.57 | 4.78 | ✓ | 72.25% | 2.3 | 76.8 | 21.5 | 214.8 | ✓ | 3.24 | 0.82 | 0.14 |
| **Literature** | | | | | | | | | | | | | | | | | | |
| eSEN-30M-OAM [21] | 0.925 | 0.061 | 0.018 | 0.170 | 15.00 | 4.00 | 9.13 | 5.73 | ✓ | | - | - | - | - | - | - | - | - |
| ORB v3 [58] | 0.905 | 0.075 | 0.024 | 0.210 | - | - | - | - | - | | - | - | - | - | - | - | - | - |
| SevenNet-MF-ompa [36] | 0.901 | 0.064 | 0.021 | 0.317 | - | - | - | - | - | | - | - | - | - | - | - | - | - |
| GRACE-2L-OAM [8] | 0.880 | 0.067 | 0.023 | 0.294 | - | - | - | - | - | | - | - | - | - | - | - | - | - |
| MACE-MPA-0 [6] | 0.852 | 0.073 | 0.028 | 0.412 | - | - | - | - | - | | - | - | - | - | - | - | - | - |
| eqv2-OC20 [42] | - | - | - | - | - | - | - | - | - | 60.80% | - | - | - | - | - | - | - | - |
| GemNet-OC20 [42] | - | - | - | - | - | - | - | - | - | 54.88% | - | - | - | - | - | - | - | - |
| eSEN-sm-cons. [43] | - | - | - | - | - | - | - | - | - | | 4.52 | 147.3 | 28.6 | 268.6 | ✓ | - | - | - |
| eSEN-S-OMC [23] | - | - | - | - | - | - | - | - | - | | - | - | - | - | - | 6.18 | 0.74 | 0.18 |

## E.1 Additional UMA Results

In this section, we provide results for the UMA, UMA-1, and UMA-1.1 models. The original UMA models (S, M, L) were trained on a preview subset of the OMol25 dataset ($\approx 70\%$), since portions of the dataset were still being calculated when UMA model training began. Both UMA-1 and UMA-1.1 were trained on the full OMol25 as released 05/14/2025. The other datasets used for training were unchanged, with the exception of the sampling ratios (Table 8) where the OMat24 coeffcient was changed from 4 to 2 for both UMA-1 and UMA-1.1. UMA-1.1 resolved a size extensive bug later discovered. These models were fine-tuned on our 1.0 models with the fix in place. Tables 11 and 12 correspond to Tables 2 and 3 in the main text. Additional UMA 1 series models will be included in the arXiv version of the paper.

## E.2 Materials

Table 13: Materials validation and test evaluations from OMat24 [5] and HEA. All energies are in meV, forces are in meV/Å and stresses are in meV/Å$^3$.

| Model | Val [5] | | | | Test | | | | | | | | | | | |
| | Energy/Atom | Forces | Stress | Force Cosine | WBM [5] | | | OOD Composition [5] | | | OOD Element [5] | | | HEA | | |
| | | | | | Energy/Atom | Forces | Stress | Energy/Atom | Forces | Stress | Energy/Atom | Forces | Stress | Energy/Atom | Forces | Stress |
| **UMA** | | | | | | | | | | | | | | | | |
| UMA-S | 11.3 | 57.1 | 2.9 | 0.98 | 20.0 | 60.8 | 4.4 | 11.5 | 57.0 | 3.0 | 9.9 | 56.9 | 2.6 | 22.0 | 72.8 | 3.1 |
| UMA-M | 10.0 | 47.3 | 2.7 | 0.99 | 18.1 | 51.4 | 4.3 | 10.2 | 47.6 | 2.9 | 8.5 | 47.1 | 2.4 | 19.0 | 62.2 | 3.2 |
| UMA-L | 9.7 | 43.5 | 2.5 | 0.99 | 17.6 | 45.5 | 3.8 | 9.8 | 43.6 | 2.6 | 8.1 | 43.6 | 2.3 | 24.8 | 48.3 | 2.8 |
| **Literature** | | | | | | | | | | | | | | | | |
| eSEN-30M-OMat [21] | 10.7 | 47.3 | 2.6 | 0.99 | 16.2 | 49.6 | 4.1 | 10.7 | 47.3 | 2.8 | 9.0 | 47.2 | 2.3 | 20.0 | 59.5 | 3.2 |
| eqV2-86M-OMat [44] | 10.0 | 44.9 | 2.4 | 0.99 | 14.9 | 46.3 | 3.6 | 10.0 | 44.5 | 2.5 | 8.8 | 44.7 | 2.1 | 20.3 | 47.0 | 2.7 |

Table 14: **Materials evals results.** We note that UMA models are fine-tuned on the MPTrj [17] and sAlex [61, 5] datasets to be consistent with the DFT settings of the benchmarks.

| Model | Matbench [59] | | | | | | | Kappa 103 [56] | | MDR Phonons [45] | | | | | Elasticity [16, 34] | | Binary Elasticity | | NVE MD [21] |
| | F1 | DAF | Precision | Accuracy | MAE [eV/atom] | RMSE [eV/atom] | $r^2$ | $\kappa_{sre}$ | $\kappa_{srme}$ | $\omega_{max}$, MAE [K] | $\omega_{avg}$, MAE [K] | Entropy, MAE [kJ/Kmol] | $C_v$, MAE [kJ/Kmol] | Energy, Free MAE [kJ/mol] | $G_{vrh}$, MAE [GPa] | $K_{vrh}$, MAE [GPa] | $G_{vrh}$, MAE [GPa] | $K_{vrh}$, MAE [GPa] | Conservative |
| **UMA** | | | | | | | | | | | | | | | | | | | |
| UMA-S | 0.92 | 6.00 | 0.92 | 0.97 | 0.02 | 0.07 | 0.87 | 0.09 | 0.20 | 17.59 | 7.41 | 13.59 | 3.49 | 5.00 | 8.54 | 4.96 | 8.57 | 5.25 | ✓ |
| UMA-M | 0.93 | 6.08 | 0.93 | 0.98 | 0.02 | 0.07 | 0.87 | 0.09 | 0.20 | 13.91 | 5.11 | 9.63 | 2.66 | 3.39 | 8.40 | 4.76 | 7.07 | 4.75 | ✓ |
| UMA-L | 0.93 | 6.09 | 0.93 | 0.98 | 0.02 | 0.07 | 0.86 | 0.45 | 0.67 | 78.50 | 27.68 | 43.04 | 15.85 | 18.20 | 20.56 | 14.48 | 21.95 | 17.01 | ✗ |
| **Literature** | | | | | | | | | | | | | | | | | | | |
| eSEN-30M-OAM [21] | 0.93 | 6.07 | 0.93 | 0.98 | 0.02 | 0.07 | 0.87 | - | 0.17 | 15.00 | 10.21 | 10.00 | 3.00 | 4.00 | 9.13 | 5.73 | 9.02 | 5.73 | ✓ |
| eqV2-86M-OAM [5] | 0.92 | 6.05 | 0.92 | 0.98 | 0.02 | 0.07 | 0.85 | 1.82 | 1.94 | 840.33 | 377.96 | 426.79 | 102.72 | 251.14 | 19.60 | 26.25 | 22.02 | 26.50 | ✗ |

Full results on materials' benchmarks are in Tables 13 and 14. Table 13 shows both validation and test results following OMat24 [5] along with the new high entropy alloy HEA test introduced in this paper. The HEA dataset contains relaxation trajectories for over 5000 alloys with atomic configuration disorder. Input structures were generated by sampling metallic element combinations of up to 6 different unique elements and using the special quasirandom structure method [80, 4] to decorate face-centered cubic, body-centered cubic and hexagonal close packed structures. DFT relaxations were carried out following the settings used in the OMat24 dataset [5].

In Table 14 we show full results for Matbench discovery [59], MDR phonon, elastic tensors, high entropy alloy IS2RE and NVE MD conservation benchmarks. The Matbench-Discovery benchmark evaluates a model's ability to predict ground-state thermodynamic stability by optimizing geometry and predicting energy. The thermal conductivity prediction task demands accurate modeling of harmonic and anharmonic phonons, which are crucial for precise predictions of thermal transport. The MDR Phonon benchmark assesses a model's performance in predicting phonon and vibrational thermodynamic properties. The MP elastic constant benchmark tests a model's accuracy in predicting

bulk and shear moduli, requiring precise calculations of stress tensors and their derivatives with respect to cell deformations.

## E.3 Catalysis

For catalysis, we show full validation and test results for OC20 [11] in Table 15. The structures in the dataset contain molecules, called adsorbates, interacting with surfaces. The goal is to estimate the adsorption energy, which is the change in energy as the adsorbates come into contact with the surface, and the forces on the atoms. Force MAEs are comparable across UMA-M and UMA-L to other state-of-the-art models. Adsorption energies for UMA are calculated by subtracting two total energy calculations as described in the main text, which improves results over prior models.

In addition to energy and forces MAEs, we use AdsorbML to evaluate the performance on the practically relevant task of finding the global minima adsorption energy [42]. One limitation of the originally proposed AdsorbML pipeline is that it requires a DFT evaluation of the ML-identified global minima structure. To make this benchmark more accessible to those without access to DFT, we propose a slightly altered version of this benchmark that only requires ML. Here, we follow the same AdsorbML pipeline but allow the use of the ML-predicted global minima energy, but success is now only considered if the energy is within 0.1 eV of the DFT minima. Previously, success did not enforce a lower bound so long a DFT evaluation confirmed that energy prediction is realized. Without DFT, we also set a lower bound of 0.1 eV to provide some flexibility to finding a better global minima than DFT. We show that metrics are still highly correlated when you perform the original evaluation to the one proposed here.

Table 15: Catalysis validation and test results on OC20 [11] metrics. All energies are in meV and forces are in meV/Å.

|  | Val (Total Energy) | | | | | | Test (Ads. Energy) | | | |
|  | ID | | | OOD-Both | | | ID | | OOD-Both | |
| Model | Energy | Forces | Force Cosine | Energy | Forces | Force Cosine | Energy | Force | Energy | Force |
|---|---|---|---|---|---|---|---|---|---|---|
| **UMA** | | | | | | | | | | |
| UMA-S | 63.6 | 24.1 | 0.63 | 107.0 | 29.2 | 0.65 | 52.1 | 24.3 | 70.2 | 30.9 |
| UMA-M | 43.1 | 15.8 | 0.73 | 70.0 | 19.2 | 0.75 | 33.4 | 16.0 | 46.5 | 21.0 |
| UMA-L | 32.6 | 12.0 | 0.77 | 49.8 | 14.5 | 0.79 | 32.4 | 12.2 | 43.5 | 15.9 |
| **Literature** | | | | | | | | | | |
| eqV2-OC20 [44] | - | - | - | - | - | - | 149.1 | 11.63 | 306.5 | 15.74 |
| GemNet-OC20 [22] | - | - | - | - | - | - | 163.5 | 16.33 | 343.3 | 23.11 |

## E.4 Molecules

Table 16: Open Molecule 2025 [43] validation evaluations across biomolecules, electrolytes, metal complexes, neutral organics and OOD-comp. All energies are in meV and forces are in meV/Å. All models are trained with preview OMol25.

|  | Biomolecules | | Electrolytes | | Metal Complexes | | Neutral Organics | | OOD-Comp | |
| Model | Energy/Atom | Force | Energy/Atom | Force | Energy/Atom | Force | Energy/Atom | Force | Energy/Atom | Force |
|---|---|---|---|---|---|---|---|---|---|---|
| **UMA** | | | | | | | | | | |
| UMA-S | 0.53 | 5.69 | 2.69 | 11.65 | 4.63 | 37.85 | 1.00 | 13.15 | 3.62 | 12.02 |
| UMA-M | 0.44 | 3.95 | 2.28 | 10.21 | 3.60 | 28.81 | 0.68 | 7.00 | 3.21 | 9.90 |
| UMA-L | 0.33 | 2.90 | 1.13 | 4.52 | 3.35 | 24.85 | 0.65 | 5.02 | 2.39 | 5.83 |
| **Baseline** | | | | | | | | | | |
| eSEN-S-OMol | 0.54 | 6.06 | 2.52 | 12.63 | 4.27 | 37.30 | 0.84 | 13.00 | 3.69 | 12.78 |

We report results on molecules following the Open Molecules 2025 [43]. These include energy and force estimates for validation and test splits, as well as, numerous other tasks. The validation and test results are shown in Tables 16 and 17, respectively. Note that the eSEN-S-OMol model is only trained on the preview OMol25 dataset, which is $\approx 70\%$ of the full OMol25 dataset for a fair

Table 17: Open Molecule 2025 [43] test evaluations across biomolecules, electrolytes, metal complexes, neutral organics and OOD-comp, metal ligand, PDB-TM, reactivity, COD and anions. All energies are in meV and forces are in meV/Å. All models are trained with preview OMol25.

| Model | Biomolecules Energy/Atom | Biomolecules Force | Electrolytes Energy/Atom | Electrolytes Force | Metal Complexes Energy/Atom | Metal Complexes Force | Neutral Organics Energy/Atom | Neutral Organics Force | OOD-Comp Energy/Atom | OOD-Comp Force | Metal Ligand Energy/Atom | Metal Ligand Force | PDB-TM Energy/Atom | PDB-TM Force | Reactivity Energy/Atom | Reactivity Force | COD Energy/Atom | COD Force | Anions Energy/Atom | Anions Force |
|---|---|---|---|---|---|---|---|---|---|---|---|---|---|---|---|---|---|---|---|---|
| **UMA** | | | | | | | | | | | | | | | | | | | | |
| UMA-S | 0.51 | 5.70 | 3.80 | 13.95 | 3.07 | 33.56 | 1.49 | 20.33 | 3.64 | 10.80 | 1.23 | 17.71 | 0.88 | 16.12 | 4.82 | 61.80 | 2.92 | 29.82 | 0.66 | 10.85 |
| UMA-M | 0.42 | 3.89 | 3.29 | 13.19 | 2.40 | 24.85 | 0.98 | 10.83 | 3.26 | 9.09 | 0.99 | 12.04 | 0.69 | 10.37 | 3.88 | 47.75 | 2.19 | 20.11 | 0.50 | 8.03 |
| UMA-L | 0.34 | 2.92 | 1.41 | 5.42 | 2.47 | 21.67 | 1.03 | 6.91 | 2.33 | 5.19 | 1.05 | 10.00 | 0.81 | 8.76 | 3.93 | 41.97 | 2.57 | 15.44 | 0.62 | 5.90 |
| **Baseline** | | | | | | | | | | | | | | | | | | | | |
| eSEN-S-OMol | 0.52 | 6.06 | 3.52 | 15.23 | 2.86 | 33.30 | 1.35 | 20.06 | 3.67 | 11.56 | 1.18 | 17.49 | 0.79 | 14.11 | 4.89 | 61.16 | 2.93 | 24.12 | 0.47 | 10.38 |

comparison with the UMA models that were trained on the same subset. In general, the UMA-S and eSEN-S-OMol models provide comparable results.

Table 18: Open Molecule 2025 [43] single point evaluations for protein-ligand, IE/EA, spin gap and distance scaling. All energies are in meV and forces are in meV/Å. All models are trained with preview OMol25.

| Model | Protein-ligand Ixn Energy MAE | Protein-ligand Ixn Forces MAE | IE/EA Δ Energy MAE | IE/EA Δ Forces MAE | IE/EA Δ Forces cosine sim. | Spin gap Δ Energy MAE | Spin gap Δ Forces MAE | Spin gap Δ Forces cosine sim. | Distance scaling Δ Energy (SR) MAE | Distance scaling Δ Forces (SR) MAE | Distance scaling Δ Energy (LR) MAE | Distance scaling Δ Forces (LR) MAE |
|---|---|---|---|---|---|---|---|---|---|---|---|---|
| **UMA** | | | | | | | | | | | | |
| UMA-S | 150.25 | 5.09 | 336.16 | 80.60 | 0.77 | 665.75 | 112.09 | 0.69 | 67.60 | 4.11 | 432.14 | 5.54 |
| UMA-M | 89.69 | 4.06 | 236.76 | 66.28 | 0.81 | 547.73 | 98.15 | 0.74 | 41.60 | 3.86 | 588.74 | 8.73 |
| UMA-L | 71.68 | 2.27 | 244.23 | 57.18 | 0.82 | 568.36 | 93.09 | 0.73 | 16.55 | 2.03 | 246.10 | 2.34 |
| **Baselines** | | | | | | | | | | | | |
| eSEN-S-OMol | 154.48 | 5.59 | 310.19 | 77.48 | 0.77 | 634.02 | 110.28 | 0.70 | 73.02 | 4.16 | 608.91 | 7.69 |

Table 19: Open Molecule 2025 [43] optimization evaluations including ligand-strain, conformer prediction, and protonation states. Results are reported across a variety of energy and structure based metrics for each task. All energies are in meV. All models are trained with preview OMol25.

| Model | Ligand strain Strain energy MAE [meV]↓ | Ligand strain RMSD min. [Å]↓ | Conformers RMSD ensemble [Å]↓ | Conformers RMSD boltz. [Å]↓ | Conformers Δ Energy MAE [meV]↓ | Conformers RMSD reopt. [Å]↓ | Conformers Δ Energy reopt. [meV]↓ | Protonation RMSD ensemble [Å]↓ | Protonation Δ Energy MAE [meV]↓ | Protonation RMSD reopt. [Å]↓ | Protonation Δ Energy reopt. [meV]↓ |
|---|---|---|---|---|---|---|---|---|---|---|---|
| **UMA** | | | | | | | | | | | |
| UMA-S | 4.39 | 0.28 | 0.06 | 0.05 | 5.76 | 0.02 | 3.06 | 0.13 | 40.42 | 0.04 | 18.35 |
| UMA-M | 2.45 | 0.19 | 0.04 | 0.03 | 3.19 | 0.01 | 1.47 | 0.08 | 24.08 | 0.02 | 10.99 |
| UMA-L | 3.37 | 0.25 | 0.05 | 0.05 | 4.97 | 0.01 | 2.27 | 0.11 | 30.31 | 0.02 | 12.54 |
| **Baselines** | | | | | | | | | | | |
| eSEN-S-OMol | 5.15 | 0.31 | 0.06 | 0.05 | 5.25 | 0.03 | 3.24 | 0.13 | 32.82 | 0.04 | 18.65 |

OMol25 [43] provides numerous other tasks for evaluation. These include evaluations which only require the estimation of a single point DFT calculation and evaluations that require optimizations. The comparison for single point DFT calculations is shown in Table 18. Similar to the test results, the UMA-S and eSEN-S-OMol models perform similarly with the UMA-M and UMA-L performing significantly better. Table 19 shows the results for tasks that require optimizations, which require repeated calls to the model to update atoms positions until a local energy minima is found. The small models report similar performance, and the UMA-M demonstrates the highest accuracies. It is likely that UMA-M outperforms UMA-L due to UMA-M being energy conserving and better behaved during optimization tasks.

Table 20: Open Molecular Crystals 2025 [23] validation and test table. All energies are in meV, forces are in meV/Å and stresses are in meV/Å$^3$.

| | Val | | | | Test | | | |
| Model | Energy/Atom | Forces | Stress | Force Cosine | Energy/Atom | Forces | Stress | Force Cosine |
|---|---|---|---|---|---|---|---|---|
| **UMA** | | | | | | | | |
| UMA-S | 0.9 | 4.9 | 1.0 | 0.92 | 0.9 | 4.8 | 1.0 | 0.93 |
| UMA-M | 0.8 | 3.1 | 1.0 | 0.95 | 0.8 | 3.0 | 1.0 | 0.95 |
| UMA-L | 0.6 | 2.3 | 0.1 | 0.96 | 0.6 | 2.3 | 0.1 | 0.96 |
| **Baselines** | | | | | | | | |
| eSEN-S-OMC | 1.06 | 5.58 | 0.96 | 0.92 | 1.05 | 5.39 | 0.94 | 0.92 |

Table 21: Open Molecular Crystals 2025 [23] evaluation for polymorphs from the Structure Ranking Phase of CCDC 7th CSP Blind Test [29]. All lattice energies are per molecule basis in kJ/mol.

| | CCDC 7th CSP Blind Test Polymorphs | | | | | | | |
| Model | Lattice Energy MAE [kJ/mol] | RMSE [kJ/mol] | $r^2$ | Rank correlation Kendall | Spearman | Structures RMSD [Å] | RMSD sd [Å] | Match rate |
|---|---|---|---|---|---|---|---|---|
| **UMA** | | | | | | | | |
| UMA-S | 2.69 | 3.67 | 0.73 | 0.82 | 0.93 | 0.12 | 0.07 | 0.99 |
| UMA-M | 2.66 | 3.71 | 0.60 | 0.81 | 0.91 | 0.13 | 0.07 | 0.99 |
| UMA-L | 2.49 | 3.70 | 0.81 | 0.84 | 0.95 | 0.12 | 0.07 | 1.00 |
| **Baselines** | | | | | | | | |
| eSEN-S-OMC | 6.18 | 7.38 | 0.07 | 0.74 | 0.87 | 0.18 | 0.08 | 0.91 |

## E.5 Molecular Crystals

Open Molecular Crystals (OMC25) [23] evaluates whether a model can predict the packing of molecule into crystal structures. This task requires the accurate estimation of inter-molecular forces. Results on validation and test splits are shown in Table 20. It is notable that all sizes of UMA models outperform the eSEN-S-OMC model trained on only OMC25. This indicates that the other datasets, such as OMol25, provide useful complementary information for the task. One important and real-world task for molecular crystals is to predict the lowest energy packing, called a polymorph, for a molecule. Results for this task for a subset of molecular crystal polymorphs from the most recent 7th Crystal Structure Prediction (CSP) Blind Test [29] are shown in Table 21. The pymatgen's [52] StructureMatcher class with default settings is used to match DFT and UMA-relaxed polymorphs, and root mean square deviation (RMSD) is computed for matches. Similar to the test metrics, the UMA models outperform the eSEN-S-OMC trained on only OMC25.

## E.6 Metal–Organic Frameworks

Table 22: OpenDAC [67] val and test table. All energies are in meV and forces are in meV/Å.

| | Val (Total Energy) | | | Test (Ads. Energy) | | | | | |
| | | | | ID | | | OOD-L/T | | |
| Model | Energy | Forces | Force Cosine | Energy | Forces | Force Cosine | Energy | Forces | Force Cosine |
|---|---|---|---|---|---|---|---|---|---|
| **UMA** | | | | | | | | | |
| UMA-S | 60.4 | 5.9 | 0.82 | 169.5 | 16.7 | 0.63 | 292.4 | 16.0 | 0.57 |
| UMA-M | 59.3 | 3.8 | 0.91 | 167.3 | 14.8 | 0.62 | 290.2 | 10.7 | 0.76 |
| UMA-L | 38.7 | 3.3 | 0.91 | 177.1 | 7.8 | 0.82 | 291.1 | 6.5 | 0.91 |
| **Literature** | | | | | | | | | |
| eqV2-ODAC [67] | - | - | - | 145.0 | 8.2 | 0.69 | 316.0 | 7.2 | 0.72 |

The results on OpenDAC [67] are shown in Table 22. The OpenDAC dataset contains Metal-Organic Frameworks (MOFs) with $CO_2$ and water molecules. The goal is to estimate the change in energy in the presence with and without the $CO_2$ and water molecules. These adsorption energies are computed in the same manner as for catalysts. The use of total energies leads to significantly better adsorption

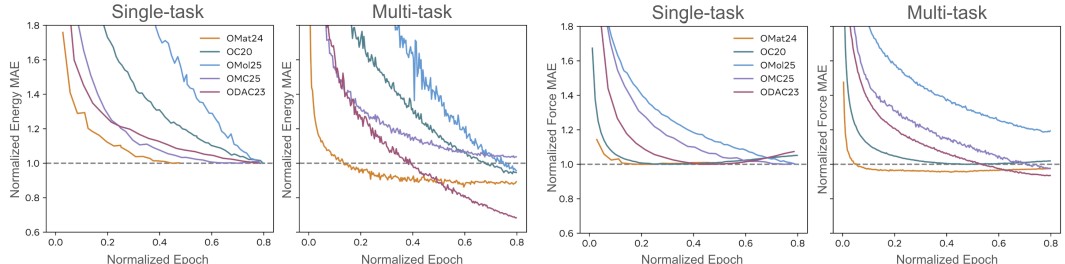

Figure 4: Pre-training curves of UMA-L for both single-task and multi-task models. Errors are normalized based on single-task performance. Note single-task models can overfit (forces on right), and the multi-task model generally converges to lower errors.

energy estimates, similar to catalysis. The forces of UMA-M and UMA-L are similar to the SOTA eqV2-ODAC [67] model.

## F   Single-task vs Multi-task

For large models, multi-task training offers benefits even without MoLE. In Figure 4, we plot the direct-force pre-training curves of UMA-lg and single-task models with the same model architecture and size. All metrics are normalized to those achieved with the models trained on single tasks to easily compare their relative performance with multi-task models. We observe that single task models frequently overfit to forces (OMat overfits upon further training), while the multi-task UMA model does not. Furthermore, UMA achieves lower losses in most cases. The one exception is OMol forces, for which errors are already small (< 10 meV/Å) for both models.

## G   Additional MoLE Analysis

### G.1   Expert Analysis

Using a limited validation set consisting of 10,000 OMat24, 5,000 OC20, 20,000 OMol25, 10,000 OMC25, and 5,000 ODAC23 samples, we calculate the mean expert coefficient for each element-expert pair across all systems where the pair appears. We visualize these results using 32 periodic tables, each representing one of the 32 experts in UMA-S (Figure 5).

Additionally we visualize the expert cofficient mean and variance across datasets (tasks). Many experts utilized in OC20 are also used in OMat24. In contrast, the experts associated with OMol25 show minimal overlap with other datasets, aside from a small subset shared with OM25C. Lastly ODAC23 and OMC25 utilize the fewest experts, and these two share a single expert between them.6).

### G.2   Generalization Across Architectures

Table 23: Testing the generalization capability of MoLE layers. We applied 8-expert MoLE layers to both eSEN[21] and EquiformerV2[44] model architectures and measured their relative performance on direct pretraining against the versions without no MoLE. For each row, we show the relative improvement over the baseline.

| OMol25 Val subset | eSEN 6M | EquiformerV2 6M |
|---|---|---|
| Metal Complexes | +10% | +10% |
| Spice | +5% | +5% |
| Neutral organics | 0% | +3% |
| Biomolecules | +25% | +21% |
| Electrolytes | +12% | +10% |
| **Mean across subsets** | **+10%** | **+10%** |

We performed additional analysis to ablate the benefit of the MoLE layers. Since our MoLE implement is very general and can be applied any linear layers inside a model, we ran experiments adding MoLE

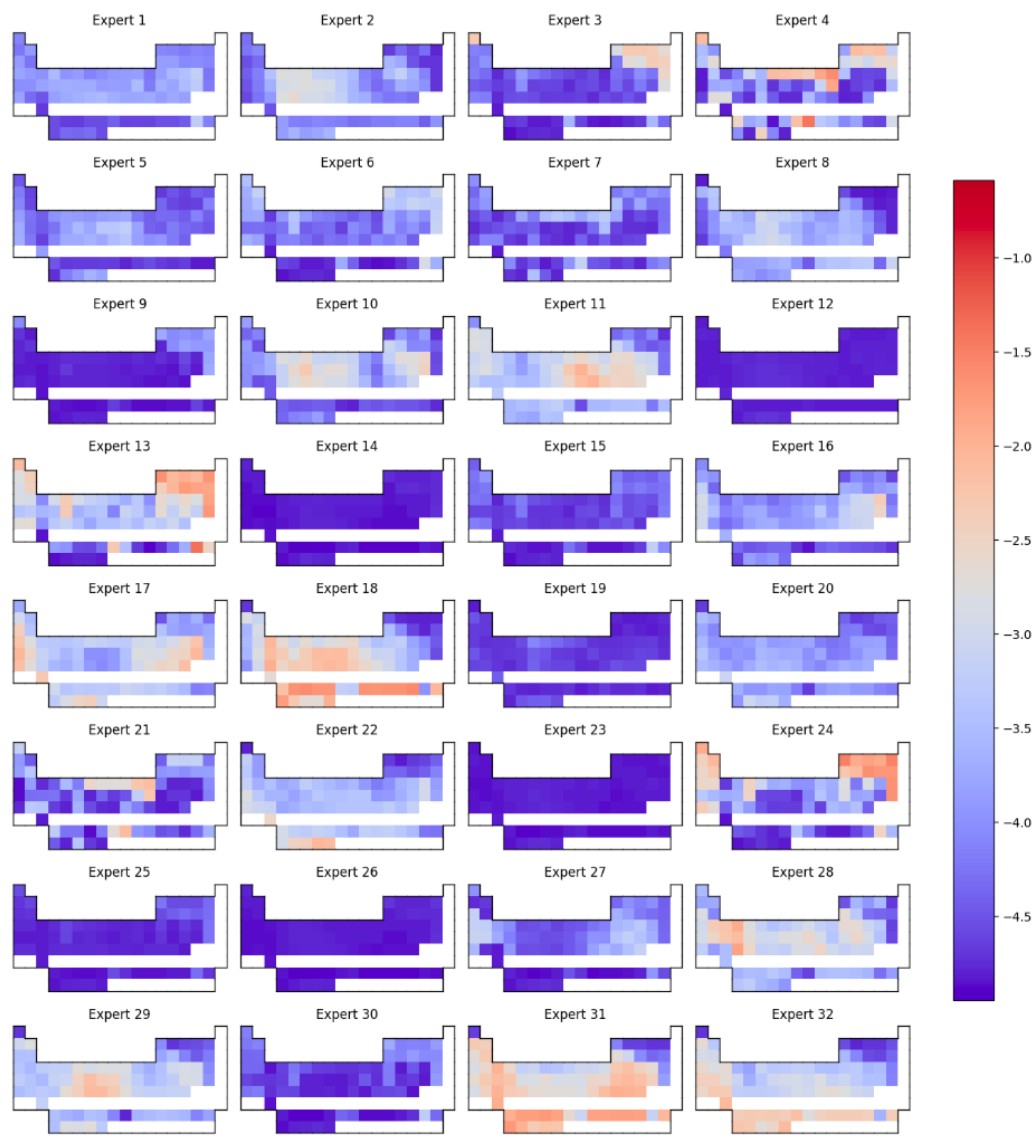

Figure 5: Log mean expert coefficient across element-expert pairs.

to both eSEN and EquiformersV2 architectures, controlling for model size (6M base parameters) and 8 training epochs on Omol dataset, direct pretraining only.

23 shows that despite EquiformersV2 and eSEN being very different architectures, the relative benefit of adding MoLE is fairly consistent in both cases. This suggests that MoLE is a simple and general approach to boost the capacity of MLIP models without incurring inference speed and memory overhead.

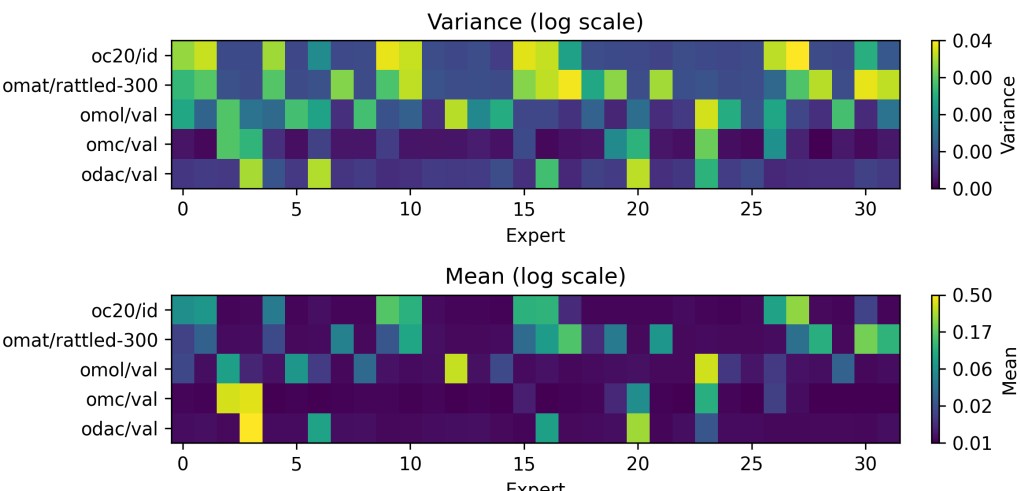

Figure 6: Log mean expert coefficient across element-expert pairs.

