# OpenReview forum: "UMA: A Family of Universal Models for Atoms"
_NeurIPS.cc/2025/Conference — NeurIPS 2025 spotlight_

### Official Review · Reviewer_WjpQ · 2025-06-23

**Clarity:** 3
**Significance:** 3
**Originality:** 4
**Rating:** 5
**Confidence:** 4

**Summary:**

The paper introduce a new set of models, Universal Models for Atoms (UMA), for atomistic structures spanning periodic crystals, molecules and surfaces interacting with molecules. In addition to the new models and architectures, the paper also outlines a study of scaling laws for atomistic models that aims to provide insight on the optimal relationship between data size, model size and compute budget. The paper starts by motivating the need for Machine Learning Interatomic Potentials (MLIPs) to accelerate the computation of accurate energies and forces of atomic systems based on Density Functional Theory (DFT) data. The introduction also poses the question of whether MLIPs can be trained across different types of DFT types and whether scaling of data and model sizes can unlock breakthroughs for atomistic modeling with machine learning. The proposed UMA models are trained on a combination of five datasets spanning 500M atomic systems and include scaling laws for different size UMA models, including UMA-sm, UMA-md and UMA-lg.

Section 2 describes the overall approach starting with the UMA architecture. The UMA architecture is based of eSEN, which leverages message passing in spherical harmonic embeddings, and is augmented by global information that can represent particularities of the how the underlying DFT data was obtained (e.g., charge and spin). Furthermore, UMA uses a mixture of linear expert (MoLE) that combine experts based on the sum of a set of weights multiplied by a contribution. The paper claims that the MoLE approach enables smooth variation of prediction while also maintaining rotational equivariance. The training procedure undergoes two stages: The first stage performs direct force prediction and the second stage without a force head that uses conservative energy predictions via autograd. Given the large of training, the paper outlines additional efficiency improvements, including: pretraining with BF16 followed by FP32 fine-tuning and amortization of GPU memory usage for the MoLE layers. Section 3 briefly describe the datasets used for training UMA models.

Section 4 describes the main results focusing on model and data scaling, optimal model size and modeling performance. The scaling laws analysis based on a fixed compute FLOP budget provides new insights into large-scale MLIP training, including a power law for UMA model training that serves the size of the different models. The results also include a discussion on overfitting challenges during training leading to deviations from the fitted power law. The scaling also leads to the observation that MoLE can be more parameter efficient compared to a dense model, thereby enabling more efficient inference for downstream applications. The second set of results compared single task and multi-task performance with MoLE models. The results in Table 2 generally show competitive performance of the UMA models while single task models still outperform in their specific task. The most compelling results for UMA are shown in Table 3 where the models show outperformance across different tasks and benchmarks for the atomistic systems covered in the training set. Section 5 provides a discussion of related work on MLIPs and scaling relation highlighting the need to use different DFT settings for different systems and the utility of scaling laws for model design. The paper ends with a discussion of limitations focusing on the limitations of the cutoff distance and the need to explore better ways to embed relevant DFT parameters into the model input for greater generalizability.

**Questions:**

* Can you describe in more detail why MoLE preserves rotational equivariance?
* When you say pretraining happens with BF16 does that refer to the direct force pretraining or to both stages?
* Can you describe why UMA-lg is not conservative?
* Where the models in Table 3 fine-tuned for the task in the benchmark? Can you provide more details on the performance difference between Table 2 and Table 3?

**Ethical Concerns:**

["NO or VERY MINOR ethics concerns only"]

**Final Justification:**

I maintain my support and original score as the authors clarified many of details asked in my review. While some questions remain unresolved (e.g., scaling applicability to other methods), they constitute future work in my opinion.

**Limitations:**

yes

**Quality:**

4

**Strengths And Weaknesses:**

Overall, the paper provides a valuable contribution to the field of MLIPs with new, performant model architectures and insights on scaling laws for data, model size and compute.

**Strengths:**
* The paper provides a novel, performant model architecture that combines atomistic data from different sources and includes a way to different the source of different DFT calculations. The architecture shows promising scalability for downstream applications given the MoLE design. [Originality, Significance]
* The study of scaling laws represents a useful insight for the MLIP domain given that few studies have provided this. [Significance, Quality]
* The paper presents a compelling case for scaling MLIPs with more and lays out the methods and experiments with good clarity and relevant detail. [Clarity]

**Weaknesses:**
* While the paper has a strong technical foundation, it could provide a more detailed analysis on the consequences and limitations of its scaling study [Significance]. Some questions to consider:
    * Do the authors expect this type of scaling to hold for other types of architectures or only for UMA?
    * How important is the MoLE to the observed scaling compared to the eSCN convolution? This could have consequences for future model design. The dense vs MoLE isoflops provide some good data for this, so a better discussion would be interesting.
    * Given the increased scale, how important are engineering optimizations to achieve greater scale? Are there additional challenges, such as mitigating overfitting?

---

> ### Author Rebuttal · Authors · 2025-07-31
>
> We would like to thank reviewer WjpQ for their time and the constructive feedback! We address each of the reviewer’s comments/questions below.
>
> **Questions**
>
> >Q1: Can you describe in more detail why MoLE preserves rotational equivariance?
>
> Our router is invariant to rotations as it only depends on the reduced atomic composition (not atomic positions), global charge/spin, and the task. Adding MoLE swaps one linear operation with another linear operation during the SO2 convolution. Since these linear operations are used in the same manner as the original SO2 convolution, the block maintains its equivariance as described in the eSCN paper.
>
> >Q2: When you say pretraining happens with BF16 does that refer to the direct force pretraining or to both stages?
>
> The direct pre-training stage utilizes BF16 whereas the conservative fine-tuning stage uses FP32.
>
> >Q3: Can you describe why UMA-lg is not conservative?
>
> We opted to make UMA-lg a direct model because it can still run inference on a single 80GB GPU for smaller systems i.e. < 1k atoms. In contrast, a conservative model at that size would require model parallelism, making it too computationally expensive for most of the MLIP community. Additionally, the extra training overhead associated with conservative models (~3.5x) is impractical at the lg model size.
>
> >Q4: Where the models in Table 3 fine-tuned for the task in the benchmark? Can you provide more details on the performance difference between Table 2 and Table 3?
>
> The UMA models in Table 3 were fine-tuned for the materials evaluations not for catalysis, molecules, or molecular crystal evaluations. The reason the models were fine-tuned (on MPtraj and sAlex) for the materials evaluations is to ensure a consistent level of theory (DFT settings). UMA models are trained on OMat24 data which has a slightly different level of theory compared to the Matbench Discovery benchmark and other materials evaluations that are common in the field. In practice, we recommend using the OMat24 level of theory (i.e. the standard UMA OMat task), so this is just an issue for evaluation. None of the models in Table 2 are fine-tuned, this was for materials evaluations in Table 3.
>
> **Perceived weaknesses**
>
> >PW1: Do the authors expect this type of scaling to hold for other types of architectures or only for UMA?
>
> We believe it is likely that the log-linear scaling behavior holds for a larger family of models due to UMA’s similarity to other architectures that exhibit such scaling behaviors (Kaplan et al., 2020, Brehmer et al., 2024). In particular since UMA’s learnable parameters almost entirely reside in linear layers, akin to LLMs, UMA’s training compute (flops) scales linearly with the number of parameters N and size of dataset D, where a LLM token can be thought as an edge in UMA (see Appendix C). There are also other studies in equivariant GNNs that show this behavior (e.g. Brehmer et al., 2024). However, we do not have enough evidence currently to support or reject this hypothesis. We hope this encourages more independent scaling law studies in this space in the future to better understand these questions!
>
> > PW1 cont: How important is the MoLE to the observed scaling compared to the eSCN convolution? This could have consequences for future model design. The dense vs MoLE isoflops provide some good data for this, so a better discussion would be interesting.
>
> We believe MoLE is independent of the scaling behavior compared to the eSCN convolution. In Figure 3c-e, we found the power laws coefficients with and without MoLE look similar (with the main distinction being a constant offset). The offset can be explained by the fact that MoLE just gives the eSCN convolution a reservoir of additional parameter capacity (~2.5x effective parameter increase from Figure 3e) at a cost of more memory and compute (which can be alleviated by pre-merging the MoLE experts during inference).
>
> >PW1 cont: Given the increased scale, how important are engineering optimizations to achieve greater scale? Are there additional challenges, such as mitigating overfitting?
>
> In order to scale effectively, we need to optimize both for training time and GPU-memory.
> Many optimizations are important to reduce training time across all scales such as two-stage direct/conserved training, mixed BF16/FP32 precision etc.
>
> For memory, different optimizations need to be applied depending on the dominant consumer of GPU memory. There are typically 2 main types of memory consumers:
> - Parameters - when total parameter count is close or >1B, they start to become a significant bottleneck on 80GB GPUs (optimizer states, EMA, other copies etc can take up 10s GBs quickly), parameter sharding techniques like FSDP are important to enable further scaling
> - Activations - activations scale with the number of layers, amount of operations and number of input atoms/edges and atom representation sizes. To perform conservative batched training with large systems, we need techniques like activation checkpointing and graph parallelism to reduce and distribute activation memory across GPUs. For example, conservative fine-tuning with Uma-md was very difficult without graph parallelism.
>
> Furthermore, dynamic batching in MLIP training with a large variety of system sizes can make training very inefficient (due to frequent allocations and memory fragmentation) and lead to OOMs. We found our max-atom batching scheme was critical to make sure training is memory stable and efficient throughout the run.
>
> We observed overfitting frequently when training large models (ie: UMA-lg) on smaller or less diverse datasets (Section F of the Appendix). However, we found when training on large mixed datasets (i.e. the full UMA training set), it took much longer to observe overfitting and when it occurred the effect was much less pronounced. We also add expert dropout and standard regularization to help mitigate overfitting.

---

### Official Review · Reviewer_iEKm · 2025-06-30

**Clarity:** 4
**Significance:** 4
**Originality:** 3
**Rating:** 6
**Confidence:** 5

**Summary:**

This paper present a “Universal Model for Atoms” with major contributions in both pre-training model architecture and scaling of datasets to create an efficient yet highly performant series of models. The architecture is based on Mixture of Experts approaches; however, rather than standard sparse selection of experts, the authors introduce the MoLE (Mixture of Linear Experts). Unlike language models, MLFF present a regression task where the output must vary smoothly, especially with respect to expert selection. The author therefore argue for a weighted contraction of the expert weight matrices. This leads to a “fewer” number of activated parameters by the effective contraction of the weight matrix. Note, this is particularly promising for inference procedures where the contraction can be performed ahead of time and repeated through the course of the network. For training, UMA introduces a number of new techniques, including low precision training (bfloat16) and following by increasing the precision in a second stage. For datasets, UMA enables the ability to combine a number of previously independent datasets including OC20, ODAC23, and OMC25 covering an extremely wide variety of atomic interactions. Other novel architectural components are the inclusion of charge and spin within the resulting DFT task.

In the results section, the authors discuss how model and data scaling laws were used to design levels of sparsity and scale models. The Iso-FLOP training curves in particular are particular helpful for determining an optimal model given a fixed compute budget. The authors then explore how UMA—as a multi-task model—compares to models trained on individual datasets. The multi-task model (UMA) often remains competitive with top single-task models and can even improve performance (for example, on OMol) — highlighting the benefits from the transfer learning. For standard benchmarks such as Matbench-Discovery, MDR phonon, and others, UMA achieve state of the art performance. The authors highlight that this strong performance is across Materials, Catalysis, Molecules, and Molecular Crystals — indicating the generality of the methods.

**Questions:**

Questions:
1) From the appendix, it appears that the many of the experts have a very low element-expert pair utilization (expert 19, 23, and so on).  Were there load-balancing losses that were tried in order to motivate expert utilization? Similarly, the expert analysis in appendix G highlights how certain experts are correlated to certain elements of the periodic table. However, a core focus of the paper is on multi-task learning. Was similar analysis performed for each of the datasets — e.g. whether certain datasets re-use similar experts or choose different ones?
2) Is there a specific linear operation within the SO2 convolution that is replaced by the MoLE method or all of them? In particular, the integration of MoLE into the SO2 convolution is barely discussed in the core body of the paper [mainly the caption of Figure 2]. Additional details here would be appreciated. Moreover, there is a specific linear operation in the node-based feed forward as well. Was there an ablation for this network layer as well?
3) How do the speeds compare between the two stages of pretraining? Are there applications where the model at the end of stage 1 is possible to use or is the bfloat16 degradation 20%+ across the board?

**Ethical Concerns:**

["NO or VERY MINOR ethics concerns only"]

**Final Justification:**

The authors have answered my questions in the review and have made the commitment to add the details to the final version of paper. I have read through the other reviews and despite minor limitations in the form style of benchmarking (i.e. specific to two model families), I believe the questions brought up by other reviewers has been well-answered.

Highlights from the paper
1) A universal model that covers a wide variety of materials
2) An inference-time speed up with details about the implementation
3) Empirical understanding of scaling laws

Highlights from the rebuttals:
1) Discussion around load-balancing and shared experts between datasets highlights interpretability
2) Training innovations clearly stated

All of these points are positives for the paper, so I recommend acceptance.

**Limitations:**

Yes

**Paper Formatting Concerns:**

Not applicable.

The additional anonymous variant of the paper does not follow Neurips formatting but the provided paper and appendix in the supplementary material do.

**Quality:**

3

**Strengths And Weaknesses:**

Strengths:
1) The paper introduces a highly ‘universal’ MLIP that is able to combine a wide variety of datasets into a single model. The universality of this model, from materials, catalysis, molecules, molecular crystals, to ODAC (MoFs) means that this model can have wide-scale applicability. In particular, the transfer learning between different domains (e.g. with the improvements in Molecular Crystals which seems to benefit from multi-task learning) highlights how the model has enough capacity to learn the best of each dataset.
2) The introduction of an inference-optimized MLIP that can be run efficiently. In particular, the idea of contracting weight ahead of time — and amortizing this cost over the simulation — is novel to this application and seems to yield very promising results. The UMA method introduces a new way to scale MLIPs in a way that will remain optimized for inference.
3) The paper is extremely well-written with a number of ablations that really help understand the benefits of the UMA model. For example, the scaling laws in terms of the number of active parameters really help show how the sparse models improve the performance for a fixed number of parameters. Also, the IsoFLOP curves help understand how compute should be split to train similar models under fixed compute budgets. All of these detailed findings provide a very explicit understanding of the models and what can be discovered.

Limitations:
1) The Mixture-of-Linear Experts (MoLE) method [to the best of my understanding] has been introduced previously in Mixture of Experts literature. For example, the method is highly similar to dense mixture of expert strategies — with particular similarity to SMEAR [https://arxiv.org/pdf/2306.03745] and Lory [https://arxiv.org/pdf/2405.03133]. Both also incorporate the idea of a pre-merged FFN for inference procedures of a learned router. This does not detract significantly from the originality of this work as I believe that it is the first of its-kind to apply this method to graph networks. However, it would be worth mentioning the other method works and potentially toning down the claims of originality.
2) The implementation of UMA method in training and inference is insufficiently discussed in the core body of the paper. For example, standard issues such as expert capacity would likely be of concern when the data points of the atomistic systems are of variable sizes. The inference speeds of the methods suggest some form of acceleration but details of the implementation as well as further exploration into the efficiency would be worth discussing.

---

> ### Author Rebuttal · Authors · 2025-07-30
>
> We would like to thank reviewer iEKm for their time and the constructive feedback! We address each of the reviewer’s comments/questions below.
>
> **Questions**
>
> >Q1: From the appendix, it appears that the many of the experts have a very low element-expert pair utilization (expert 19, 23, and so on). Were there load-balancing losses that were tried in order to motivate expert utilization? Similarly, the expert analysis in appendix G highlights how certain experts are correlated to certain elements of the periodic table. However, a core focus of the paper is on multi-task learning. Was similar analysis performed for each of the datasets — e.g. whether certain datasets re-use similar experts or choose different ones?
>
> We did not try load balancing, but you’re right there are a few experts with low utilization and it's possible this could be improved. This experiment will take a bit of time but we will try to include a preliminary result (to inform future work, we won’t retrain all models) in the appendix for the camera-ready version of the paper.
>
> Based on your comment, we conducted a comparable analysis focusing on the expert weightings across different datasets (i.e. tasks). We describe it here in words per NeurIPS rules on images, but we will include the figure in the expert analysis section of the appendix. The results indicate that OC20, OMat, and OMol employ the largest number of experts. Notably, many experts utilized in OC20 are also used in OMat. In contrast, the experts associated with OMol show minimal overlap with other datasets, aside from a small subset shared with OMC. Lastly, the datasets ODAC and OMC utilize the fewest experts, and these two share a single expert between them.
>
> >Q2: Is there a specific linear operation within the SO2 convolution that is replaced by the MoLE method or all of them? In particular, the integration of MoLE into the SO2 convolution is barely discussed in the core body of the paper [mainly the caption of Figure 2]. Additional details here would be appreciated. Moreover, there is a specific linear operation in the node-based feed forward as well. Was there an ablation for this network layer as well?
>
> The SO2 convolution consists of 1 linear layer for M=0 and two linear layers for other values of M. We replace each one of these linear layers with a MoLE. In fact our code is written in a general way that searches inside specific modules and then replaces linear layers with MoLE layers. We do this for all SO2 convolution operations in the network (2 x Num layers). We do not replace the linear operations inside the node-based feed forward layers. Given the camera-ready version can be 1 page longer than the original submission we can expand the description of MoLE in the main text.
>
> >Q3: How do the speeds compare between the two stages of pretraining? Are there applications where the model at the end of stage 1 is possible to use or is the bfloat16 degradation 20%+ across the board?
>
> The conservative fine-tuning stage is much slower than the direct pre-training stage, \~3.5x slower than the UMA-sm model. The BF16 degradation is fairly universal, but this can be fixed with a short amount of FP32 fine-tuning (~ 50k steps). After the small amount of FP32 fine-tuning the models would be usable but they would be direct force models and these have a number of known limitations.
>
> **Perceived weaknesses**
>
> >PW1: The Mixture-of-Linear Experts (MoLE) method [to the best of my understanding] has been introduced previously in Mixture of Experts literature. For example, the method is highly similar to dense mixture of expert strategies — with particular similarity to SMEAR [https://arxiv.org/pdf/2306.03745] and Lory [https://arxiv.org/pdf/2405.03133]. Both also incorporate the idea of a pre-merged FFN for inference procedures of a learned router. This does not detract significantly from the originality of this work as I believe that it is the first of its-kind to apply this method to graph networks. However, it would be worth mentioning the other method works and potentially toning down the claims of originality.
>
> Our understanding is inline with yours that MoLE-like strategies have been previously discussed in the MoE literature. We included the citations we knew about on line 122, but we agree with the ones you mentioned as well. We will include these citations in the camera-ready version of the paper.
>
> > PW2: The implementation of UMA method in training and inference is insufficiently discussed in the core body of the paper.
>
> We agree, however, due to the 9-page limit of the NeurIPS format, we had to move most of the technical details of training and inference to the appendix. Given we get an extra page for the camera-ready version of the paper we can try to incorporate more of these details into the main text.
>
> >PW2 cont: For example, standard issues such as expert capacity would likely be of concern when the data points of the atomistic systems are of variable sizes.
>
> The routing of MoLE experts in our implementation does not depend on the size of the atomic system, only on charge, spin, task, and composition. Furthermore, the composition embedding is based on the reduced atomic composition. For example a system with 1 carbon and 100 carbons would see the same composition embedding and therefore the same expert routing.
>
> >PW2 cont: The inference speeds of the methods suggest some form of acceleration but details of the implementation as well as further exploration into the efficiency would be worth discussing.
>
> We discussed briefly in supplementary materials but we agree with the reviewer that we should add more details to the final paper. In particular, there were significant innovations used to speed up the model and improve its memory efficiency.
>
> 1. Chunked activation checkpointing
>
>     - During inference with a large number of atoms, the activations used in our SO2 convolutions dominate GPU-memory as they scale linearly with the number of atoms, edges and embedding size. Standard activation checkpointing only provides slight improvements to memory because it still scales with the number of atoms. However, if we break up or "chunk" the input and compute them sequentially during SO2 convolutions at inference time and checkpoint the chunked activations, this allowed us to scale UMA-sm from 2k atoms to 100k with just ~30% inference time penalty.
>
> 2. Pre-merge of MoLE experts
>
>     - This has been discussed as one of the central themes of MoLE. It allows us to skip the router call and expert parameter memory entirely and turn UMA into eSEN essentially at inference time. This is particularly important for large models like UMA-md where the parameter count can be brought down from 1.4B to 50M, dramatically reducing the footprint during inference time.
>
> 3. Specialized GPU graph construction
>
>     - We have an in-house GPU graph construction algorithm that makes generating graphs for large systems fast. We evaluated many 3rd systems such as Pymat Gen, cuML KNN, openMM NNOps etc and none of them was fast enough or as simple to use for the scale that we want.
>
> 4. Other optimizations
>
>      - Based on profiling, we eliminated many inference time bottlenecks that made eSEN particularly slow such as inefficient slicing operations, un-fused operators etc. We also added optimizations like cuda-graphs for Wigner matrix generation (collapsed many small operations), TF32 and torch compile to get further speed-ups

---

> ### Comment · Reviewer_iEKm · 2025-08-05
> **Discussion**
>
> I have read through all of the reviewers comments as well as the rebuttals from the authors.
>
> My review was focused mainly on additional details from individual sections (e.g. router utilization), and the answers in rebuttals suggest that these details will be added to the main body of the paper. In addition, the inference speedups as well as custom implementations (i.e. of graph creation) seem like impressive technical improvements. In particular, an only 30% slowdown at inference seems like a significant win -- allowing for drastically different systems in a single setting.
>
> I maintain my support for the paper and keep my score/confidence.

---

### Official Review · Reviewer_isJH · 2025-07-02

**Clarity:** 3
**Significance:** 4
**Originality:** 2
**Rating:** 5
**Confidence:** 4

**Summary:**

This paper introduces a family of foundation machine learning (ML) potential models, called Universal Models for Atoms (UMA). These models are trained on a large-scale dataset that combines multiple publicly available materials datasets, encompassing a wide range of atomic structures.

The main characteristics of the model are as follows:
* It adopts a mixture of linear experts architecture, inspired by the mixture of experts technique used in large language models.
* It introduces a two-stage training procedure designed to improve the overall training efficiency: the first stage predicts forces and stresses directly, and the second stage fine-tunes the model using autograd to produce energy-conserving outputs.
* It takes total charge and total spin multiplicity as additional inputs, enabling a potential extension to more diverse quantum chemical settings.

The paper presents extensive experiments on prediction accuracy, inference efficiency, and empirical scaling laws. Moreover, the full details of the model, including hyperparameters, code, and pretrained checkpoints, are publicly released, ensuring reproducibility.

**Questions:**

**Question:**

I am curious about the intended directions for future extensions of this work. Will the authors prioritize expanding the range of properties that the model can predict (e.g., spectra, magnetism), or will they focus on further scaling up the dataset used in training? In the latter case, extending the model to all-atom protein structures also seems like a promising possibility.

**Suggestion 1:**

In the ML potential field, the evaluation of model quality is gradually moving beyond simple energy and force prediction accuracy toward more sophisticated assessments of stability and reliability. Several recent works have proposed additional evaluation metrics. It would be valuable for the authors to consider including some of these metrics in their evaluation. Belows are relevant references for such discussions:
* Raja et al., “Stability-Aware Training of Machine Learning Force Fields with Differentiable Boltzmann Estimators.” TMLR (2025).
* Yi et al., “Towards Physically Reliable Molecular Representation Learning.” UAI (2023).
* Fu et al., “Forces are not Enough: Benchmark and Critical Evaluation for Machine Learning Force Fields with Molecular Simulations.” TMLR (2023).

**Suggestion 2 (minor):**

In Figure 3, it would be better to align the y-axis ranges for panels (a) and (b), as well as for (c) and (d). While the current setup does not hinder interpretation of general trends, it makes direct comparisons more difficult.

**Ethical Concerns:**

["NO or VERY MINOR ethics concerns only"]

**Final Justification:**

After reviewing the authors' rebuttal and considering other reviewers' comments along with the authors' responses to them, I find that the authors have adequately addressed my concerns. Therefore, I maintain my original assessment.

**Limitations:**

Yes

**Quality:**

3

**Strengths And Weaknesses:**

**Strengths:**
* This paper presents a unified ML potential model that covers a wide range of material types, including molecules and crystals. By releasing both the checkpoints and code, the work makes a significant contribution to the field.
* The paper conducts an empirical analysis of scaling laws using such a large-scale model, which is an experiment that most researchers would find difficult to carry out. This provides valuable insights into this domain.

**Weaknesses:**

While the scope of materials handled by the baseline models may vary, the current manuscript omits quite a few results in the comparison tables. More comprehensive baseline comparisons would strengthen the paper.

**Recommendation:**

Although the comparison with baseline models is somewhat limited, the release of a foundational ML potential model applicable across diverse material structures, along with the scaling law analysis, represents a substantial contribution to the field. In light of this, I recommend accepting this paper.

---

> ### Author Rebuttal · Authors · 2025-07-31
>
> We would like to thank reviewer isJH for their time and the constructive feedback! We address each of the reviewer’s comments/questions below.
>
> **Questions**
>
> >Q1: I am curious about the intended directions for future extensions of this work. Will the authors prioritize expanding the range of properties that the model can predict (e.g., spectra, magnetism), or will they focus on further scaling up the dataset used in training? In the latter case, extending the model to all-atom protein structures also seems like a promising possibility.
>
> We are quite excited about the future of these types of models! One of the benefits of open sourcing the work is that the community can build off of it and extend our work. There is a lot of opportunity to fine-tune our models to predict other properties such as band gaps and for distillation to generate specialized models that are even faster. We are particularly excited about the possibility of multi-GPU inference, as mentioned in the paper, that enables much larger simulations such as you mentioned for all-atom protein structures.
>
> As for scaling up the dataset, it will likely be more targeted in the future. One of the exciting observations from this work is that in a number of cases the models are already at or very close to chemical accuracy so scaling further isn’t needed. That is not true across the board and there are clearly areas that would benefit from more data. Understanding what data is needed and what data has the highest information content will be the focus of future work.
>
> >Q2: In the ML potential field, the evaluation of model quality is gradually moving beyond simple energy and force prediction accuracy toward more sophisticated assessments of stability and reliability. Several recent works have proposed additional evaluation metrics. It would be valuable for the authors to consider including some of these metrics in their evaluation. ...
>
> We agree there is a lot of value in assessing downstream evaluations in addition to the standard energy and force metrics. In our paper, Table 2 focuses on the standard energy and force metrics whereas Table 3 focuses on downstream evaluations that require multiple energy/force evaluations. To highlight a few:
>
> - kSMRE (thermal conductivity), which requires both second and third order derivatives of the potential computed through finite differences (Póta et al., 2024).
> - AdsorbML, global (not just local) optimization problem for adsorption energy (Lan et al., 2023).
> - NVE Molecular Dynamics (MD) energy drift test, this evaluation requires running 100 ps of NVE MD while monitoring the energy drift, which is a proxy for the smoothness of the MLIP (Fu et al., 2025). If the total energy drift is below a defined threshold, we say the model is conservative. UMA-lg does not pass this test because it is a direct force model.
>
> >Q3: In Figure 3, it would be better to align the y-axis ranges for panels (a) and (b), as well as for (c) and (d). While the current setup does not hinder interpretation of general trends, it makes direct comparisons more difficult.
>
> Thanks for this feedback. In the camera-ready version we will make sure any common axes (e.g. val loss) use the same scale.
>
> **Perceived weaknesses**
>
> >PW1: While the scope of materials handled by the baseline models may vary, the current manuscript omits quite a few results in the comparison tables. More comprehensive baseline comparisons would strengthen the paper.
>
> We tried our best to include baseline results where they existed and in cases where there were none we trained our own. In Tables 2 and 3 you see a bunch of blanks “-” for baseline models because we only evaluated single-task baselines on the task they were trained on.
>
> Are there specific baselines that have been reported in the literature you think we missed? We would be happy to include them in the camera-ready version of the paper.
>
> We acknowledge that there is substantial room for improvement in benchmarking these kinds of universal MLIPs both in terms of the depth and breadth of evaluations, but in this work we went well beyond what any single paper has done in the past in terms of benchmarks, marking a significant effort to understand model performance. In some cases we created our own benchmarks as part of this work e.g. HEAs for materials.

---

> > ### Comment · Reviewer_isJH · 2025-08-05
> >
> > I have read the authors' response and the discussions with other reviewers. The authors have provided sufficient responses, and most of my questions have been addressed.
> >
> > Regarding the baseline experiments, I initially considered requesting results from retraining or fine-tuning existing models on each individual dataset, such as fine-tuning the MACE-OFF model for the case of Table 3. However, I have reconsidered this position, recognizing that the experiments conducted in this paper could be viewed as establishing a new benchmark for universal machine learning potentials that encompasses all types of materials.
> >
> > Based on these considerations, I maintain my initial rating.

---

### Official Review · Reviewer_tgt6 · 2025-07-03

**Clarity:** 3
**Significance:** 3
**Originality:** 2
**Rating:** 4
**Confidence:** 4

**Summary:**

This paper presents UMA, a novel family of Universal Models for Atoms, aiming to achieve state-of-the-art speed, accuracy, and generalization in atomic simulations across diverse chemical domains. The authors leverage an exceptionally large dataset (half a billion 3D atomic structures) and introduce a Mixture of Linear Experts (MoLE) architecture to efficiently scale model capacity. The work demonstrates that a single UMA model can perform comparably to or better than specialized models without fine-tuning across various applications in chemistry and materials science.

**Questions:**

1. The manuscript frequently refers to an "Appendix" (e.g., Appendix A, B.6, C, D, E, F). However, these appendices appear to be missing from the provided PDF document. Could the authors please ensure that the complete supplementary material, including all referenced appendices, is included in the final submission?
2. While the paper introduces the Mixture of Linear Experts (MoLE) architecture and applies it to the eSEN base, the novelty of MoLE is very limited.  Could the authors provide a clearer summary of the specific model or theoretical innovations developed in this paper beyond the large-scale data training? Additionally, a more comprehensive ablation study comparing MoLE against alternative architectural choices or existing MoE implementations would strengthen the claim of its efficiency and effectiveness. In its current form, the paper leans more towards a technical report on large-scale training rather than highlighting fundamental architectural breakthroughs.
3. The authors claim that UMA models are designed to push the frontier of speed, accuracy, and generalization, and that a single model can perform similarly or better than specialized models without fine-tuning. However, upon reviewing Table 2, it appears that UMA models do not consistently outperform all specialized baselines. For instance, in the "WBM Energy" column, the eqV2-OMat model (14.9 meV) shows a lower error than even the largest UMA model, UMA-lg (17.6 meV). Similar observations can be made for other metrics. Could the authors clarify these discrepancies and elaborate on the specific contexts or metrics where UMA's generalization truly translates to superior performance compared to highly specialized models?
4. Following up on Table 2, it is observed that UMA-lg, despite having the largest number of parameters, does not always exhibit the best performance among the UMA family (e.g., UMA-md often performs comparably or better). This seems to deviate from the expected behavior suggested by the empirical scaling laws presented in the paper (e.g., Figure 3). Could the authors explain why the scaling laws might not fully apply to these specific evaluation tests, or if there are other factors contributing to this non-monotonic performance with increasing model size?
5. Regarding the training data, the paper mentions combining datasets from various chemical domains. It is known that different DFT calculators and levels of theory (e.g., PBE functional with plane-wave codes for materials vs. ORCA software with Gaussian basis sets for molecules) result in different reference zero-energy lines. While an "energy referencing scheme" is briefly mentioned (line 161), but the appendix B.6 is not given in the pdf. Could the authors provide a more detailed explanation of how these disparate setups across different domains were reconciled during data compilation and model training to ensure consistency and prevent performance degradation?
6. The "Related Work" section (Section 5, starting line 293) provides a good overview but appears to omit several highly relevant and recent works in the field of universal MLIPs or large-scale atomistic modeling. Specifically, works such as Orb, PET-MAD, MatterSim, DPA, and SevenNet also contribute significantly to this field. Including these references would provide a more comprehensive overview of the current landscape and better contextualize UMA's contributions.

**Ethical Concerns:**

["NO or VERY MINOR ethics concerns only"]

**Final Justification:**

Thank you for your response. Your rebuttal has addressed most of my concerns. While I still have some reservations regarding the novelty of the work, I appreciate the clarifications provided and have accordingly updated my evaluation to a borderline accept.

**Limitations:**

The authors did not discuss the limitations of their model in the paper. I would strongly suggest the authors to answer my questions above and put some of the answers as part of the limitation discussion in the main text.

**Quality:**

3

**Strengths And Weaknesses:**

* Strengths:

1. The paper addresses a critical challenge in MLIPs by aiming for a single, generalizable model across a wide range of chemical domains (molecules, materials, catalysis).
2. Very large training data size, probably largest to date.
3. The authors demonstrate scaling laws

* Weakness:

1. This paper looks like a technical report, instead of a research paper.
2. The paper only contains very limited theoretical innovation and it only trained models on top of existing architecture on a large dataset.

---

> ### Author Rebuttal · Authors · 2025-07-31
>
> We would like to thank reviewer tgt6 for their time and the constructive feedback! We address each of the reviewer’s comments/questions below.
>
> **Questions**
>
> >Q1: The manuscript frequently refers to an "Appendix" (e.g., Appendix A, B.6, C, D, E, F). However, these appendices appear to be missing from the provided PDF document. …
>
> Our appendix was included in the Supplementary Material, which is standard practice for NeurIPS and was available for reviewers to download (see link on review page). We apologize it wasn’t easier to find, but at least one other reviewer referenced data from the appendix, so it was accessible at the time of review.
>
> >Q2: While the paper introduces the Mixture of Linear Experts (MoLE) architecture and applies it to the eSEN base, the novelty of MoLE is very limited. ...
>
> Below we have listed the major novel contributions in this paper:
>
> 1. MoLE architecture
>     - To the best of our knowledge, MoLE (or any type of MoE) is a novel design choice for a top performing MLIP. Standard MoE architectures borrowed from LLMs cannot be easily applied to MLIPs as it is difficult to preserve equivariance, ensure smoothness of the potential energy surface (PES), and make them memory and compute efficient at inference time.
>
>      - Our MoLE routing is specifically designed to depend on static features of the atomic structures (charge, spin, composition, etc.) rather than dynamic latent space features compared to standard MOEs. This allows us to pre-merge our experts weights before running a sequential simulation, making it fast and memory efficient. If the routing depended on the positions of atoms it could change throughout a sequential simulation and introduce a discontinuity in the PES (see Fu et al., 2025). During development, we tested dynamic routing schemes but ultimately rejected them as they didn't meet our listed requirements.
>
> 2. Handling total charge/spin
>
>     - Most universal MLIPs e.g. MACE, Orb, Mattersim, DPA, SevenNet, PET-MAD etc. do not include charge and spin, but it is particularly important for molecules.
>
> 3. Handling different DFT levels of theory
>
>    - We are not aware of another top performing universal MLIP that can generalize to multiple diverse tasks (for example in materials + molecules space) with different DFT levels of theory without relying on fine-tuning or regressing on performance.
>
> 4. Scaling laws
>
>     - We provide the first systematic study of empirical scaling relationships including compute, data, and model size in MLIPs.
>
> 5. Training innovations
>
>     - In section A of the Appendix, we highlight several innovations that made training at this scale possible including mixed BF16 (previously never used in MLIP training)/FP32 training, max-atom batching, and max neighbor switching.
>
> A number of these contributions and their novelty were specifically noted by the other reviewers:
> - Reviewer iEKm: “The introduction of an inference-optimized MLIP that can be run efficiently. In particular, the idea of contracting weight ahead of time — and amortizing this cost over the simulation — is novel to this application and seems to yield very promising results.”
> - Reviewer wJPq: “The paper provides a novel, performant model architecture that combines atomistic data from different sources and includes a way to different the source of different DFT calculations.”
> - Reviewer isJH: “It takes total charge and total spin multiplicity as additional inputs, enabling a potential extension to more diverse quantum chemical settings.”
>
> >Q2 cont: Additionally, a more comprehensive ablation study comparing MoLE against alternative architectural choices or existing MoE implementations would strengthen the claim of its efficiency and effectiveness. ...
>
> Based on your comment, we ran additional experiments to test the generalization ability of the MoLE architecture and applied it to EquiformerV2 on the OMol25 dataset. We found that the relative energy improvement on a similar sized EquiformerV2 (6M) and eSEN (6M) with MoLE is very comparable (see table below). This gives us more confidence that MoLE is a general approach to improve model capacity for MLIPs while preserving equivariance and simulation speed. We will include the results in the appendix of the camera-ready paper.
>
> | **OMol25 Val subset** | **eSEN 6M** | **EquiformerV2 6M** |
> | :- | :- | :- |
> | Metal Complexes | +10% | +10% |
> | Spice | +5% | +5% |
> | Neutral organics | 0% | +3% |
> | Biomolecules | +25% | +21% |
> | Electrolytes | +12% | +10% |
> | **mean across subsets** | **+10%** | **+10%** |
>
> In addition, in the main text we demonstrated with ablations that naively training multi-task models in the small parameter regime without MoLE severely degrades performance (up to 20-30%) on nearly all tasks (Figure 2 top right). This provides further evidence that multi-task training is nontrivial and the MoLE architecture is a necessary component for training smaller models, which are critical to the community.
>
> >Q2 cont: In its current form, the paper leans more towards a technical report on large-scale training rather than highlighting fundamental architectural breakthroughs.
>
> While we do believe our work introduces a number of novel techniques for the model and for training/inference (as listed above), we note that this is not the only criteria for acceptance. Novel contributions related to model scaling have also been rewarded at NeurIPS, one famous example is the GPT-3 paper which won an outstanding paper award in 2020 and was focused on how scaling up LLMs improves performance without introducing any architectural updates.
>
> >Q3: The authors claim that UMA models are designed to push the frontier of speed, accuracy, and generalization ...
>
> We agree that UMA does not outperform all specialized baselines on every single metric — we tried to be transparent about this fact. However, what is impressive is that a single model is on par or in a majority of cases better than SOTA across multiple chemistry areas (materials, molecules, catalysis, etc.). We would also like to point out that no previously published result has shown that multi-task models can even perform close to on-par with single-task variants in the space of MLIPs.
>
> Note that for WBM energy, both eqv2 and UMA perform very well and are within the threshold of practical utility (10-20 meV/atom). The strength of UMA is that while it does solid on single-point energy metrics like WBM, it also performs well on physical properties like thermal conductivity (kSMRE in Table 3) where models like eqv2 cannot be used.
>
> A few examples where UMA’s generalization shines is catalysis OOD Both energy, MOFs OOD-L/T energy (the hardest OOD splits from OC20 and ODAC25 respectively), and unseen CSP target lattice energy.
>
> >Q4: Following up on Table 2, it is observed that UMA-lg, despite having the largest number of parameters, does not always exhibit the best performance among the UMA family …
>
> We expect the empirical scaling laws to hold around the order of magnitudes of compute that we measured (10^18 - 10^20 flops) and similar to other scaling law studies (LLAMA3, Chinchilla, etc.) we expect our results to extrapolate out 2-3 orders of magnitude beyond what we measured.
>
> The deviation of UMA-lg’s performance from other models can be explained from several confounding factors:
>
>
> - UMA-lg is a direct-force model and is generally not comparable with the performance of UMA-sm/md which are gradient-force models. There can be quantitative differences between direct-force and gradient-force models, for example direct models tend to do better in single-point calculations (Table 2) while gradient-force models perform better with longer rollout tasks or ones that depend on having smooth gradients with respect to the energy (Table 3). Ideally, we would show the conserving lg model performs better however it is too computationally expensive.
> - The scaling laws we measured here are based on validation loss which only provides a rough guiding metric for model performance in the MLIP space. There are many examples where improving validation loss does not translate to improved downstream performance:
>   - Benchmarks that are close to saturation, for example in Matbench Discovery F1 scores.
>   - Benchmarks that are limited by other factors such as smoothly varying potential surface, i.e. Matbench discovery kSRME metric/phonons etc.
> - We observed empirically (and noted in the main text) that scaling beyond UMA-lg with MoLE or increased dense parameters (up to 2-10B parameters) leads to very little further improvement even in validation loss which suggests that at these model sizes we are limited by the current dataset size and diversity. We did not have the compute budget to further explore this question in this paper.
>
> >Q5: Regarding the training data, the paper mentions combining datasets from various chemical domains. ...
>
> As mentioned above, we included the appendix B.6 (Referencing) in the supplementary material. To briefly answer your question here: we reference all energies to isolated atomic energies computed with the respective level of theory (corrected by the heat of formation), followed by a linear referencing scheme (described in B.6) to help normalize the energies. This may help align the zero-energy line but importantly it enables the model to make predictions in a similar numerical range which helps with stability during training. We give the model a task embedding that allows the model to learn the energy landscape given the level of theory of the training data.
>
> >Q6: Related work ...
>
> Thanks for the suggestion, we will include these references for the camera-ready version of the paper.
>
> **Limitations**
>
> We did provide a limitations paragraph in the main text of the paper (see lines 319-328), this was acknowledged by all the other reviewers. Given your comments/suggestions, we plan to add additional content on the limitations of scaling laws to the camera-ready version of the paper.

---

> > ### Comment · Reviewer_tgt6 · 2025-08-06
> >
> > Thank you for your response. Your rebuttal has addressed most of my concerns. While I still have some reservations regarding the novelty of the work, I appreciate the clarifications provided and have accordingly updated my evaluation to a borderline accept.

---

### Note · Authors · 2025-08-16

We would like to thank the reviewers and the area chair for their time and effort in evaluating our work. We are encouraged by all the positive feedback, with reviewers recognizing the significance and impact of our work (tgt6, isJH, iEKm, WjpQ), the originality of our approach (isJH, iEKm, WjpQ), the scaling law analysis (tgt6, isJH, iEKm, WjpQ), the strong results and ablations (tgt6, iEKm, WjpQ), and the clarity and quality of the paper (isJH, iEKm, WjpQ). We appreciate the constructive suggestions and will incorporate all feedback and additional experiments into the final version.

---

### Decision · Program_Chairs · 2025-09-17

**Decision:**

Accept (spotlight)

**Comment:**

The paper presents a family of foundation models that can be used to predict physical properties of atom structures. The models are trained on a very large-scale dataset curated by the authors by combining several earlier datasets. The proposed models exploit Mixture of Experts (MoE) techniques to increase computational efficiency. Extensive experiments illustrate the performance of the models in various tasks.

The main strengths include extensive scale of the dataset, carefully conducted experiments, (arguably minor) architectural novelties, and overall high quality of presentation. Weaknesses and room for improvement were pointed out in terms of limited theoretical novelty and acknowledgement of the limitations and scope of the scaling properties of the models, as pointed out by the reviewers.

Overall, the paper was found to be highly relevant to a critical problem in atom-level physical modeling applications and a major step forwards in practical application of foundation models for atom structures in physics,

The authors were able to respond to most of the critical comments and questions posed by the reviewers in a satisfactory manner.